# DROP-UPCYCLING: TRAINING SPARSE MIXTURE OF EXPERTS WITH PARTIAL RE-INITIALIZATION

**Taishi Nakamura**[1,2,3]**, Takuya Akiba**[2]**, Kazuki Fujii**[1]**, Yusuke Oda**[3]**,
Rio Yokota**[1,3]**, Jun Suzuki**[4,5,3]
[1]Institute of Science Tokyo, [2]Sakana AI, [3]NII LLMC, [4]Tohoku University, [5]RIKEN
`taishi@rio.scrc.iir.isct.ac.jp, jun.suzuki@tohoku.ac.jp`

## ABSTRACT

The Mixture of Experts (MoE) architecture reduces the training and inference cost significantly compared to a dense model of equivalent capacity. Upcycling is an approach that initializes and trains an MoE model using a pre-trained dense model. While upcycling leads to initial performance gains, the training progresses slower than when trained from scratch, leading to suboptimal performance in the long term. We propose *Drop-Upcycling* – a method that effectively addresses this problem. Drop-Upcycling combines two seemingly contradictory approaches: utilizing the knowledge of pre-trained dense models while statistically re-initializing some parts of the weights. This approach strategically promotes expert specialization, significantly enhancing the MoE model's efficiency in knowledge acquisition. Extensive large-scale experiments demonstrate that Drop-Upcycling significantly outperforms previous MoE construction methods in the long term, specifically when training on hundreds of billions of tokens or more. As a result, our MoE model with 5.9B active parameters achieves comparable performance to a 13B dense model in the same model family, while requiring approximately 1/4 of the training FLOPs. All experimental resources, including source code, training data, model checkpoints and logs, are publicly available to promote reproducibility and future research on MoE.

| | **Weights** | `huggingface.co/collections/llm-jp/`
`drop-upcycling-674dc5be7bbb45e12a476b80` |
|---|---|---|
| | **Data** | `gitlab.llm-jp.nii.ac.jp/`
`datasets/llm-jp-corpus-v3` |
| | **Code** | `github.com/Taishi-N324/Drop-Upcycling` |
| | **Logs** | `wandb.ai/taishi-nakamura/Drop-Upcycling` |

## 1 INTRODUCTION

Large-scale language models (LLMs) have achieved remarkable results across various natural language processing applications (Brown et al., 2020; Wei et al., 2022; Ouyang et al., 2022; OpenAI, 2024). This success largely depends on scaling the number of model parameters, the amount of training data, and computational resources (Kaplan et al., 2020; Hoffmann et al., 2022), which leads to substantial training and inference costs of LLMs. Building and deploying high-performance models also require enormous resources, posing a significant barrier for many researchers and practitioners.

The *Mixture of Experts* (MoE) architecture has emerged as a promising approach to address the escalating resource demands of LLMs. MoE introduces multiple experts into some parts of the network, but only a subset is activated at any given time, allowing the model to achieve superior performance with reduced training and inference costs (Shazeer et al., 2017; Lepikhin et al., 2021; Fedus et al., 2021). In fact, cutting-edge industry models like Gemini 1.5 (Team et al., 2024) and GPT-4 (based on unofficial reports) (OpenAI, 2024) have adopted MoE, suggesting its effectiveness.

We refer to transformer-based LLMs without MoE as *dense models* and those incorporating MoE as *MoE models*. Upcycling (Komatsuzaki et al., 2023) is an approach that initializes and trains an MoE model using a pre-trained dense model, which aims to transfer learned knowledge for better

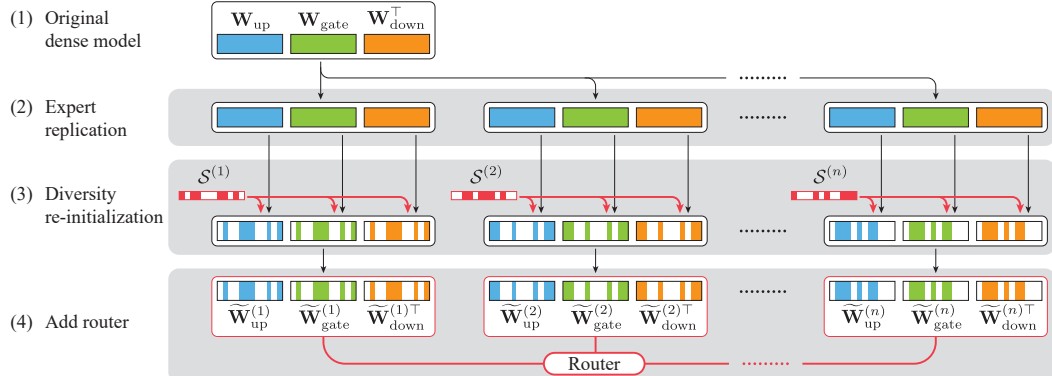

Figure 1: **Overview of the Drop-Upcycling method.** The key difference from the naïve Upcycling is Diversity re-initialization, introduced in Section 3.

initial performance. However, naïve Upcycling copies the feedforward network (FFN) layers during initialization, which makes it difficult to achieve expert specialization. This disadvantage prevents effective utilization of the MoE models' full capacity, resulting in slower convergence over long training periods. Thus, there exists a trade-off between the short-term cost savings from knowledge transfer and the long-term convergence efficiency through expert specialization.

In this paper, we propose *Drop-Upcycling* – a method that effectively addresses this trade-off, as briefly illustrated in Figure 1. Drop-Upcycling works by selectively re-initializing the parameters of the expert FFNs when expanding a dense model into an MoE model. The method is carefully designed to promote expert specialization while preserving the knowledge of pre-trained dense models. Specifically, common indices are randomly sampled along the intermediate dimension of the FFNs, and the weights are dropped either column-wise or row-wise, depending on the weight matrix types. The dropped parameters are then re-initialized using the statistics of those weights.

Extensive large-scale experiments demonstrate that Drop-Upcycling nearly resolves the trade-off between the two aforementioned challenges and significantly outperforms previous MoE model construction methods such as training from scratch and naïve Upcycling. By leveraging pre-trained dense models, Drop-Upcycling can start training from a better initial state than training from scratch, reducing training costs. On the other hand, Drop-Upcycling avoids the convergence slowdowns observed with naïve Upcycling. Specifically, in our extensive long-term training experiments, Drop-Upcycling maintained a learning curve slope similar to that of training from scratch, consistently staying ahead. This success is attributed to effective expert specialization. As a result, we constructed an MoE model with 5.9B active parameters that performs on par with a 13B dense model from the same model family, while requiring only approximately 1/4 of the training FLOPs.

This research is fully open, transparent, and accessible to all. With over 200,000 GPU hours of experimental results, conducted on NVIDIA H100 GPUs, all training data, source code, configuration files, model checkpoints, and training logs used in this study are publicly available. By providing this comprehensive resource, we aim to promote further advancements in this line of research.

Our technical contributions are summarized as follows:

- We propose Drop-Upcycling, a novel method for constructing MoE models that effectively balance knowledge transfer and expert specialization by selectively re-initializing parameters of expert FFNs when expanding a dense model into an MoE model.

- Extensive large-scale experiments demonstrate that Drop-Upcycling consistently outperforms previous MoE construction methods in long-term training scenarios.

- All aspects of this research are publicly available. This includes the MoE model with 5.9B active parameters that performs comparably to a 13B dense model in the same model family while requiring only about 1/4 of the training FLOPs.

## 2 RELATED WORK

### 2.1 MIXTURE OF EXPERTS

The concept of Mixture of Experts (MoE) was introduced about three decades ago (Jacobs et al., 1991; Jordan & Jacobs, 1994). Since then, the idea of using sparsely-gated MoE as a building block within neural network layers (Eigen et al., 2014; Shazeer et al., 2017) has evolved and has been incorporated into transformer-based language models (Lepikhin et al., 2021; Fedus et al., 2021). For a detailed overview of MoE, please refer to recent survey papers (Cai et al., 2024). Sparsely-gated MoE is currently the most common approach for building large-scale sparsely-activated models. In this paper, we focus on sparsely-gated MoE (also referred to as sparse MoE or sparsely-activated MoE), and unless otherwise specified, the term MoE refers to it.

There are various designs of MoE layers and ways to integrate them into transformer-based LLMs. For example, in addition to the standard token-centric routing, expert-centric routing has also been proposed (Zhou et al., 2022). To incorporate common knowledge, it has been suggested to introduce shared experts that are always activated (Dai et al., 2024). To simplify the discussion, we assume the most standard top-$k$ token choice routing as the MoE layer and a decoder-only transformer-based LLM that uses MoE layers only in the FFNs as the MoE model. These are common design choices for recent MoE-based LLMs, such as Mixtral (Jiang et al., 2024), Skywork-MoE (Wei et al., 2024), Phi-3.5-MoE (Abdin et al., 2024), and Grok-1[1]. Specifically, these models use 8 experts (Mixtral and Grok-1) or 16 experts (Skywork and Phi-3.5-MoE), with the top-2 experts being activated per input token. Our experiments also use top-2 routing with 8 experts per layer, as this setup aligns with those practical configurations. These facts indicate that Drop-Upcycling can be applied to most variations of MoE models. See Section 3.1 for technical details of MoE.

### 2.2 MOE MODEL INITIALIZATION

As with conventional neural networks, MoE models can be initialized randomly and trained from scratch. However, to reduce training costs, leveraging existing pre-trained dense models has become a standard approach. Below, we introduce a few methods for achieving this.

Upcycling (Komatsuzaki et al., 2023) leverages the weights of a pre-trained dense model for initializing an MoE model by initializing the experts in the MoE layer as replicas of the FFN layers in the dense model. The main advantage of Upcycling is that it boosts the model's initial performance. However, as our experiments show, MoE models initialized with Upcycling tend to have a much slower convergence, leading to suboptimal performance when trained for longer durations.

Branch-Train-MiX (BTX) (Sukhbaatar et al., 2024) is a technique where a pre-trained dense model is replicated and fine-tuned on different datasets to produce multiple distinct expert dense models. These experts are then integrated into an MoE model, followed by additional training to optimize the routers. While this method appears to ensure expert specialization by design, Jiang et al. (2024) has highlighted that the diversity achieved in this way differs from that required for MoE layer experts, leading to suboptimal performance as a result. Our experiments also show that BTX suffers from suboptimal convergence similar to those observed in Upcycling.

Concurrent with our work, the Qwen2 technical report (Yang et al., 2024) briefly suggests the use of a methodology possibly related to Drop-Upcycling in training Qwen2-MoE. Due to the report's brevity and ambiguity, it is unclear if their method exactly matches ours. Our paper offers a valuable technical contribution even if the methods are similar. The potential application of Drop-Upcycling in an advanced, industry-developed model like Qwen2-MoE that underscores the importance of further open investigation into this approach. We acknowledge the Qwen2 authors for sharing insights through their technical report.

## 3 METHOD

In this section, we explain the Drop-Upcycling method. Drop-Upcycling initializes an MoE model by utilizing a pre-trained dense model and consists of three steps:

---

[1] https://x.ai/blog/grok-os

1. **Expert Replication:** The weights of the dense model are copied to create the MoE model. All layers, except for the FFN layers, are copied directly from the dense model. The FFN layers are replaced with MoE layers, and the original FFN weights are copied to all experts within these MoE layers.

2. **Diversity Re-initialization:** In each MoE layer, a subset of the expert parameters is randomly selected and re-initialized using the original statistical information. This promotes diversity among the experts while partially retaining the knowledge of the original model, which facilitates expert specialization during subsequent training.

3. **Continued Training:** After initialization, the MoE model is trained using the standard next-token prediction loss. Optionally, a load-balancing loss, commonly applied in MoE training, can also be incorporated.

In the following, we explain the expert initialization and diversity injection processes.

## 3.1 SwiGLU and MoE Layers

We provide a brief overview of the MoE architecture. First, we review the feedforward network (FFN) layer in transformers. The SwiGLU activation function (Shazeer, 2020), now standard in state-of-the-art LLMs like LLaMA (Touvron et al., 2023) and Mixtral (Jiang et al., 2024), will be used for explanation here. However, it should be noted that Drop-Upcycling can be applied to transformers with any activation function. The FFN layer with SwiGLU is defined as follows:

$$\text{SwiGLU}(\mathbf{x}) = (\text{Swish}(\mathbf{x}^{\text{T}}\mathbf{W}_{\text{gate}}) \odot \mathbf{x}^{\text{T}}\mathbf{W}_{\text{up}})\mathbf{W}_{\text{down}}. \tag{1}$$

Here, $\mathbf{x} \in \mathbb{R}^{d_h}$ represents the input vector and $\odot$ denotes the Hadamard product. Each FFN layer contains the following three weight matrices: $\mathbf{W}_{\text{gate}}, \mathbf{W}_{\text{up}} \in \mathbb{R}^{d_h \times d_f}$, and $\mathbf{W}_{\text{down}} \in \mathbb{R}^{d_f \times d_h}$. The dimensions $d_h$ and $d_f$ are referred to as the hidden size and intermediate size, respectively.

When MoE is introduced into a transformer, each FFN layer is replaced with an MoE layer, while the rest of the architecture remains unchanged. Let us assume we use $n$ experts and Top-$k$ gating. An MoE layer comprises a router and $n$ expert FFNs. The router has a weight matrix $\mathbf{W}_{\text{router}} \in \mathbb{R}^{d_h \times n}$. The $i$-th expert FFN is denoted as $\text{SwiGLU}^{(i)}(\mathbf{x})$, which, like a standard FFN layer, consists of three weight matrices. These weights are denoted as $\mathbf{W}_{\text{gate}}^{(i)}, \mathbf{W}_{\text{up}}^{(i)}$, and $\mathbf{W}_{\text{down}}^{(i)}$. The output $\mathbf{y}$ of the MoE layer is computed as follows:

$$\mathbf{y} = \sum_{i=1}^{n} g(\mathbf{x})_i \cdot \text{SwiGLU}^{(i)}(\mathbf{x}), \tag{2}$$

where $g(\mathbf{x})_i$ is the $i$-th element of the output $g(\mathbf{x}) \in \mathbb{R}^n$ of the Top-$k$ routing function, defined as:

$$g(\mathbf{x}) = \text{Softmax}(\text{Top-}k(\mathbf{x}^{\text{T}}\mathbf{W}_{\text{router}})). \tag{3}$$

Since $k < n$ is typically the standard setting, only the top-$k$ selected experts out of $n$ are computed. Therefore, the MoE layer is sparsely activated, meaning that only a subset of the parameters is involved in the computation. The number of parameters engaged in the computation for a given input is referred to as the *active parameters* of the MoE model. This value is widely used as an approximation for the computational cost as it correlates well with the cost of both training and inference. For non-MoE models, the total number of parameters corresponds to the active parameters as all parameters are involved in every computation.

## 3.2 Expert Replication

Following (Komatsuzaki et al., 2023), we first construct a Transformer with MoE layers by replicating the weights from a pre-trained Transformer with standard FFN layers. As explained earlier, the architecture remains identical except the FFN layers, so we simply copy the weights of all non-FFN components. Each FFN layer needs to be replaced with an MoE layer, and the new MoE layers are constructed as follows: The router weights $\mathbf{W}_{\text{router}}$ are initialized randomly. For the $n$ experts, the weights from the original FFN are copied, such that $\mathbf{W}_{\text{gate}}^{(i)} = \mathbf{W}_{\text{gate}}, \mathbf{W}_{\text{up}}^{(i)} = \mathbf{W}_{\text{up}}$, and $\mathbf{W}_{\text{down}}^{(i)} = \mathbf{W}_{\text{down}}$.

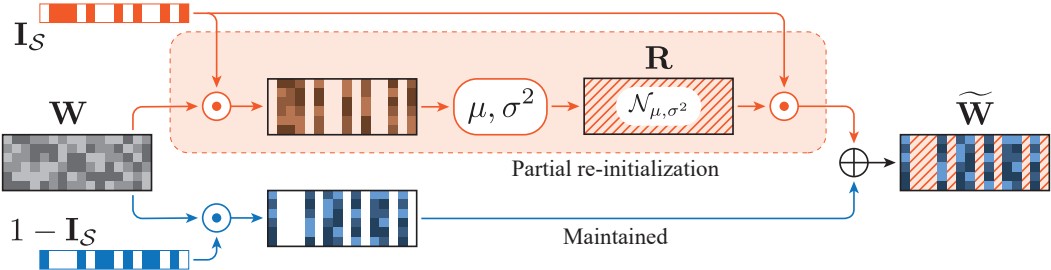

Figure 2: **Initialization of expert weights.** Columns (rows) are selected according to a set of randomly selected indices of the intermediate layer $\mathcal{S}$, then all elements of them are re-initialized with the normal distribution. Other columns (rows) are maintained.

Drop-Upcycling can also be applied to fine-grained experts and shared experts (Dai et al., 2024). See Appendix C.6 for details.

### 3.2.1 DIVERSITY RE-INITIALIZATION

Diversity re-initialization is the key step in Drop-Upcycling. This process is carefully designed to balance between knowledge retention and expert diversification. In particular, it is crucial to drop original weights along the intermediate dimension of the FFN layer based on shared indices across all three weight matrices. Specifically, the following operation is applied to every expert FFN in every MoE layer.

**Step 1: Column-wise Sampling.** We sample indices from the set of integers from 1 to intermediate size $d_f$, namely, $\mathcal{I}_{d_f} = \{1, 2, \cdots, d_f\}$, to create a set of partial indices $\mathcal{S}$. A hyperparameter $r$ ($0 \leq r \leq 1$) controls the intensity of re-initialization, determining the proportion $r$ used for sampling. That is, $\mathcal{S} \subseteq \mathcal{I}_{d_f}$ and $|\mathcal{S}| = \lfloor r d_f \rfloor$.

**Step 2: Statistics Calculation.** We calculate the mean and standard deviation of the matrices of the weights corresponding to the selected indices $\mathcal{S}$. Specifically, we compute the mean and variance $(\mu_{\text{up}}, \sigma_{\text{up}})$, $(\mu_{\text{gate}}, \sigma_{\text{gate}})$, and $(\mu_{\text{down}}, \sigma_{\text{down}})$ from the values obtained only from the non-zero columns of $\mathbf{I}_{\mathcal{S}}$ in the products $\mathbf{I}_{\mathcal{S}} \odot W_{\text{gate}}$, $\mathbf{I}_{\mathcal{S}} \odot W_{\text{up}}$, and $\mathbf{I}_{\mathcal{S}} \odot W_{\text{down}}^{\top}$, respectively, where $\mathbf{I}_{\mathcal{S}}$ is the indicator matrix whose values are 1 in the $i$-th column for $i \in \mathcal{S}$ and 0 otherwise.

**Step 3: Partial Re-Initialization.** Finally, using the calculated statistics, we perform partial re-initialization of the three weight matrices $\mathbf{W}_{\text{gate}}$, $\mathbf{W}_{\text{up}}$, and $\mathbf{W}_{\text{down}}$, obtaining $\widetilde{\mathbf{W}}_{\text{gate}}$, $\widetilde{\mathbf{W}}_{\text{up}}$, and $\widetilde{\mathbf{W}}_{\text{down}}$. For the selected indices, the weights are dropped and re-initialized randomly, while for the unselected indices, the original weights are retained.

Let $\mathbf{R}_{\text{type}}$ be a matrix whose values are sampled from the $\mathcal{N}(\mu_{\text{type}}, (\sigma_{\text{type}})^2)$ distribution, where type is one of the gate, up, or down, i.e., type $= \{\text{gate}, \text{up}, \text{down}\}$. We then obtain $\widetilde{\mathbf{W}}_{\text{type}}$ by using the following equation:

$$\widetilde{\mathbf{W}}_{\text{type}} = \mathbf{I}_{\mathcal{S}} \odot \mathbf{R}_{\text{type}} + (1 - \mathbf{I}_{\mathcal{S}}) \odot \mathbf{W}_{\text{type}}, \tag{4}$$

where we consider that the matrices, $\widetilde{\mathbf{W}}_{\text{type}}$, $\mathbf{R}_{\text{type}}$, $\mathbf{W}_{\text{type}}$ are all transposed if type $=$ down.

Figure 2 illustrates how we generate a single expert weight matrix from the original dense weights.

### 3.2.2 THEORETICAL CHARACTERISTICS

Applying the re-initialization strategy explained above, the initial MoE model obtained by Drop-Upcycling has the following characteristics:

1. **Parameter sharing among experts**: since each expert retains the original representations with a ratio $(1-r)$, with Top-k routing where $k$ experts are selected, approximately $(1-r)^k$ of representations are preserved.

2. **Characteristics of initial feedforward layers**: Consider the output of an MoE layer with parameter re-initialization ratio $r$:

$$\mathbf{y} = \text{FFN}_{\text{common}}(\mathbf{x}) + \sum_{i=1}^{N} g(\mathbf{x})_i \cdot [\text{FFN}_{\text{retained}_i}(\mathbf{x}) - \text{FFN}_{\text{common}}(\mathbf{x}) + \text{FFN}_{\text{diverse}_i}(\mathbf{x})] \quad (5)$$

where $\text{FFN}_{\text{common}}$ represents the output from parameters that are common to all selected $k$ experts (the proportion of such parameters is approximately $(1-r)^k$ due to each expert independently preserving a ratio $(1-r)$ of original parameters), $\text{FFN}_{\text{retained}_i}$ is expert $i$'s output using uniquely retained original parameters (ratio $(1-r)$), and $\text{FFN}_{\text{diverse}_i}$ is the output using reinitialized parameters (ratio $r$). The estimation error in the number of common parameters has magnitude $O\left(\frac{1}{\sqrt{d_f}}\right)$. A detailed derivation is provided in Appendix C.5.

## 4 EXPERIMENTAL SETUP

We conducted experiments to demonstrate the effectiveness of Drop-Upcycling described in Section 3. To clarify our model configurations, we introduce a notation where, for example, "8×152M" denotes an MoE model with eight experts and whose base dense model size is 152M.

We selected the Llama (Touvron et al., 2023) and Mixtral (Jiang et al., 2024) architectures for dense and MoE models, respectively, for our experiments. We employed 8 experts and the dropless (Gale et al., 2023) token choice top-2 routing (Shazeer et al., 2017) for the MoE. Detailed descriptions of the model configurations are provided in Appendix A.3

We evaluated four different methods to build MoE models, namely, training from scratch, naïve Upcycling (Komatsuzaki et al., 2023), Random Noise Upcycling (Komatsuzaki et al., 2023) and Branch-Train-MiX (Sukhbaatar et al., 2024) to compare the performance with Drop-Upcycling. Moreover, we also evaluated dense models to provide a reference of the typical performance of LLMs in our configuration and illustrate the performance gains of MoE models. We initialized all parameters of dense models using a Gaussian distribution $\mathcal{N}(0, 0.02)$. The dense models are also used as the seed models of MoE models, except when we train MoE models from scratch. When training MoE models from scratch, we used the same initialization method as the dense models, that is, $\mathcal{N}(0, 0.02)$. In Random Noise Upcycling, Drawing from (Muennighoff et al., 2024), we initialize by copying the dense model parameters and then add Gaussian noise $\mathcal{N}(0, 0.02)$ to 50% of the weights in each FFN layer. In Branch-Train-Mix, we first obtained three distinct expert dense models by further training a seed dense model with 100B extra tokens of either Japanese, English, or code. Then, we used the four dense models (the seed dense model and three expert dense models) to initialize the parameters of an MoE model. Specifically, we averaged all parameters in the four dense models except the FFN layers and duplicated the FFN layers in each model twice to build eight MoE experts. Note that this method involved extra training steps with 300B more tokens compared to the other MoE construction methods.

Unless otherwise stated, dense models were trained on 1T tokens, and MoE models were trained on 500B tokens. Our training data was obtained from publicly available data. We describe the detailed statistics of the training datasets in Appendix B.1. We followed the typical training configurations used in Llama to train dense models and Mixtral for MoE models. Details of the hyper-parameters we used are described in Appendix A.4. Moreover, the implementation and the computational environment used in our experiments are described in Appendix A.2.

We conducted a comprehensive evaluation using a wide range of tasks in Japanese and English. We used 12 evaluation datasets that can be categorized into seven types. The details of the evaluation datasets and metrics are described in Appendix B.2.

## 5 RESULTS AND DISCUSSION

In this section, we address the following questions through experiments: Is Drop-Upcycling superior to existing MoE construction methods, and does Drop-Upcycling resolve the issue of slower convergence? (Section 5.1) Does it perform well even in large-scale settings? (Section 5.2) What is the impact of the re-initialization ratio $r$? (Section 5.3) How are the experts specialized? (Section 5.4)

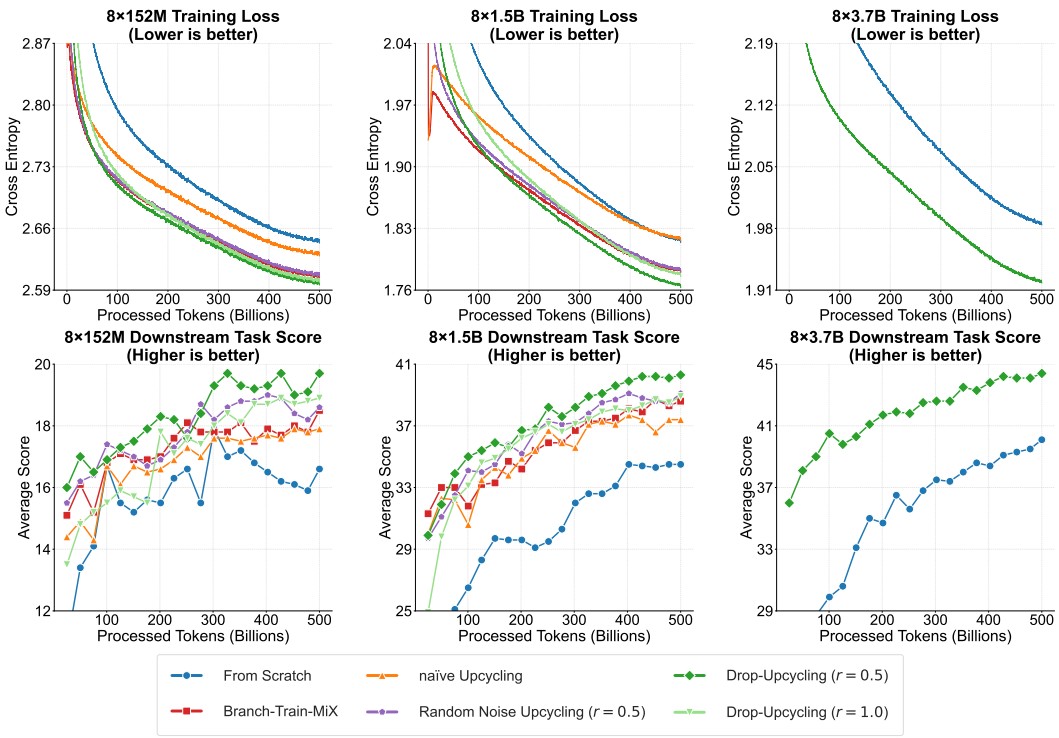

Figure 3: **Comparison of learning curves for different MoE construction methods**. The top and bottom rows illustrate the changes in training loss and downstream task scores during training, respectively. In both metrics, the proposed method, Drop-Upcycling with $r = 0.5$, achieves the best performance, gaining initial knowledge transfer while avoiding convergence slowdown.

## 5.1 METHOD COMPARISON

First, we compare Drop-Upcycling with existing methods using small (8×152M) to medium (8×1.5B) scale settings. The left two columns of Figure 3 illustrate the learning curves under these settings. The top and bottom rows illustrate the changes in training loss and downstream task scores during training, respectively. Note that in LLM pretraining, training loss serves as a reliable performance indicator since the risk of overfitting is low. The performance on downstream tasks is represented by the average score across 12 tasks, which is commonly used as the overall evaluation metric. A detailed breakdown will be discussed later in conjunction with Table 1.

Figure 3 shows that Drop-Upcycling at $r = 0.5$ (green) is significantly more efficient compared to other methods. The top row shows the training loss, while the bottom row displays the evaluation scores using downstream tasks. In both metrics and for both model sizes, Drop-Upcycling becomes the clear winner after some training. Notably, the slope of the learning curve, which indicates convergence rate, is superior. Furthermore, it can be observed that the slope of the learning curve is consistent with the case of training from scratch, suggesting that Drop-Upcycling resolves the crucial challenge of balancing knowledge transfer and expert specialization in Upcycling. For further analysis on expert specialization, see Section 5.4.

Among existing methods, naïve Upcycling exhibited the slowest loss reduction rate and improvement in task scores. Branch-Train-Mix, which starts MoE training after each expert has been trained for 100B steps on different domains such as Japanese, English, and code, initially shows an advantage over naïve Upcycling due to this favorable initialization. However, its long-term learning pace is on par with naïve Upcycling, and it is ultimately overtaken by Drop-Upcycling. As an ablation study, we evaluated setting $r = 1.0$ in Drop-Upcycling, in addition to the standard $r = 0.5$. This configuration involves random initialization of all FFNs while reusing weights for embeddings and self-attention layers. This configuration might seem inefficient at first glance. Nevertheless, our

Table 1: **Comparison of evaluation results between models with different initialization.** Training from scratch (FS), Branch-Train-Mix (BTX), naïve Upcycling (NU), Random Noise Upcycling (RNU) and Drop-Upcycling (DU) are compared. * BTX requires additional 300B tokens to obtain specialized dense models before MoE construction. Bold letters indicate the highest score within each model size.

| | Model | | Training | | Individual Scores | | | | | | | | | | | | |
|---|---|---|---|---|---|---|---|---|---|---|---|---|---|---|---|---|---|
| # | Archi-tecture | MoE Init | Tokens | FLOPs ($\times 10^{21}$) | JEM HQA | NIILC | JSQ | XL-Sum | WMT E→J | WMT J→E | OB QA | TQA | HS | SQ v2 | XW-EN | BBH | Avg |
| *Dense 152M → MoE 8×152M:* | | | | | | | | | | | | | | | | | |
| 1 | Dense | – | 1,000B | 1.59 | 17.6 | 7.9 | 10.6 | 2.4 | 0.5 | 0.5 | 14.6 | 3.0 | 28.6 | 2.0 | 60.6 | 11.5 | 13.3 |
| 2 | MoE | FS | 500B | 0.91 | 25.2 | 13.6 | 19.4 | 1.8 | 0.9 | 0.4 | 16.6 | 2.6 | 31.2 | **12.9** | 64.4 | 10.7 | 16.6 |
| 3 | MoE | BTX | 800B* | 1.39 | 28.6 | 17.1 | 26.6 | **4.3** | 2.7 | 1.1 | **18.4** | 5.1 | **32.5** | 5.3 | **65.0** | 15.9 | 18.5 |
| 4 | MoE | NU | 500B | 0.91 | 28.2 | 16.2 | 24.4 | 3.5 | 3.0 | 1.1 | 18.2 | 5.8 | 31.9 | 4.5 | 63.5 | 14.7 | 17.9 |
| 5 | MoE | RNU (r=0.5) | 500B | 0.91 | 28.6 | 17.1 | 29.4 | 3.7 | 2.3 | 1.6 | 16.8 | 5.3 | 32.0 | 4.8 | 64.5 | 17.4 | 18.6 |
| 6 | MoE | DU (r=0.5) | 500B | 0.91 | **32.2** | **18.0** | 30.6 | 3.7 | **4.7** | **2.3** | 16.8 | **6.1** | **32.5** | 6.2 | 64.2 | **19.1** | **19.7** |
| 7 | MoE | DU (r=1.0) | 500B | 0.91 | 27.2 | 16.8 | **32.5** | 4.1 | 3.7 | 1.6 | 17.0 | 5.9 | 32.4 | 4.9 | 64.8 | 15.4 | 18.9 |
| *Dense 1.5B → MoE 8×1.5B:* | | | | | | | | | | | | | | | | | |
| 8 | Dense | – | 1,000B | 11.76 | 49.6 | 42.5 | 48.1 | 11.3 | 16.8 | 8.5 | 22.2 | 23.8 | 42.9 | 16.2 | 82.5 | 25.1 | 32.5 |
| 9 | MoE | FS | 500B | 9.05 | 48.3 | 45.4 | 59.1 | 7.5 | 16.6 | 6.9 | 26.4 | 31.5 | 47.3 | 15.0 | 83.7 | 25.9 | 34.5 |
| 10 | MoE | BTX | 800B* | 12.58 | 44.3 | 51.8 | 69.4 | 11.9 | 22.4 | **12.5** | 27.8 | 39.2 | 49.7 | 18.7 | **86.4** | 28.9 | 38.6 |
| 11 | MoE | NU | 500B | 9.05 | 50.4 | 50.6 | 61.7 | 12.4 | 21.6 | 10.5 | 26.8 | 36.2 | 47.7 | 19.0 | 85.0 | 27.2 | 37.4 |
| 12 | MoE | RNU (r=0.5) | 500B | 9.05 | **53.6** | 50.5 | 71.2 | 12.3 | 22.3 | 11.7 | 26.4 | 40.0 | 49.9 | 19.1 | 84.9 | 27.5 | 39.1 |
| 13 | MoE | DU (r=0.5) | 500B | 9.05 | 51.1 | **52.3** | **72.5** | **13.7** | **22.5** | **12.5** | **30.6** | **41.3** | **50.4** | **21.2** | 86.2 | **29.1** | **40.3** |
| 14 | MoE | DU (r=1.0) | 500B | 9.05 | 52.1 | 50.9 | 68.8 | 12.3 | 21.9 | 12.4 | 25.0 | 39.1 | 49.7 | 20.6 | 86.0 | 27.9 | 38.9 |

Table 2: **Comparison between dense and MoE with large-scale configuration.** Drop-Upcycling (DU) works well even at 8×3.7B scale. The MoE model with Drop-Upcycling outperforms dense models trained with higher computational costs, demonstrating the effectiveness of Drop-Upcycling.

| | Model | | | Training | | Individual Scores | | | | | | | | | | | | |
|---|---|---|---|---|---|---|---|---|---|---|---|---|---|---|---|---|---|---|
| # | Architecture | MoE Init | Act Params / Total Params | Tokens | FLOPs ($\times 10^{22}$) | JEM HQA | NIILC | JSQ | XL-Sum | WMT E→J | WMT J→E | OB QA | TQA | HS | SQ v2 | XW-EN | BBH | Avg |
| 1 | Dense 3.7B | - | 3.7B / 3.7B | 1,000B | 2.70 | 44.5 | 47.2 | 78.8 | 12.8 | 21.4 | 15.4 | 25.0 | 33.8 | 47.3 | 23.7 | 85.9 | 28.7 | 38.7 |
| 2 | MoE 8×3.7B | FS | 5.9B / 18B | 500B | 1.98 | **53.5** | 50.8 | 69.6 | 10.4 | 20.6 | 13.9 | 29.0 | 45.8 | 51.1 | 21.1 | 87.1 | 28.1 | 40.1 |
| 3 | MoE 8×3.7B | DU (r=0.5) | 5.9B / 18B | 500B | 1.98 | 47.5 | **57.0** | 82.2 | **16.3** | **25.0** | **19.0** | 31.2 | **53.6** | **54.4** | 26.3 | **88.5** | 32.2 | 44.4 |
| 4 | Dense 13B | - | 13B / 13B | 805B | 7.43 | 47.6 | 58.3 | **85.2** | 14.1 | 24.6 | 18.3 | **31.4** | 48.6 | 53.1 | **29.3** | 88.3 | **35.2** | **44.5** |
| 5 | Dense 3.7B | - | 3.7B / 3.7B | 2,072B | 5.58 | 42.3 | 53.2 | 80.4 | 14.3 | 22.6 | 15.9 | 28.2 | 42.2 | 50.6 | 25.8 | 87.3 | 30.9 | 41.1 |

large-scale experiments reveal that even such a seemingly naive baseline can outperform naïve Upcycling in certain scenarios. For additional analysis on the impact of the $r$ value, refer to Section 5.3.

Table 1 provides a comparison of the final downstream task performance for models trained with various methods under these 8×152M and 8×1.5B settings. Model numbers refer to the leftmost column of this table. This table also includes the dense models used for upcycling. Specifically, Model 1 is the dense model used to initialize Models 3-7, and Model 8 is used to initialize Models 10-14. The proposed method, Drop-Upcycling (DU) with $r = 0.5$, consistently demonstrates superior performance across these model scales.

## 5.2 Scaling to 8×3.7B

To further evaluate the effectiveness of Drop-Upcycling in larger-scale settings and to build a practical MoE model, we conducted experiments with an 8×3.7B configuration. Due to computational resource constraints, experiments under the 8×3.7B setting were limited to training from scratch and Drop-Upcycling with $r = 0.5$.

The rightmost column of Figure 3 illustrates the learning curves under this configuration. Similar to the 8×152M and 8×1.5B settings, Drop-Upcycling significantly outperforms training from scratch. There is an initial gain in performance due to the improved initialization, and expert diversification allows the training to progress as efficiently as in the case of training from scratch, ensuring that Drop-Upcycling never gets overtaken.

Table 2 compares the models' final downstream task performance. Model numbers refer to the leftmost column of this table. Model 1 is a dense model used as a base model for the Upcycling. Models

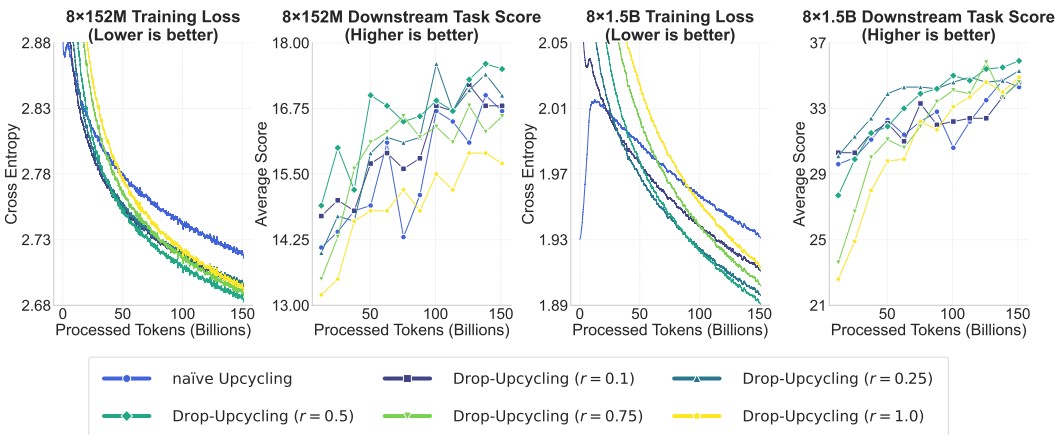

Figure 4: **Impact of re-initialization ratio** $r$**.** The training loss and downstream task score over the total number of tokens processed during training on 8×152M (left two figures) and 8×1.5B (right two figures) settings are illustrated. Even with different $r$ values, Drop-Upcycling robustly outperforms naïve Upcycling, and 0.5 appears to be the most effective ratio.

2 and 3 are MoEs built using naïve Upcycling and Drop-Upcycling, respectively, demonstrating the superiority of Drop-Upcycling. In addition, two different baseline dense models, Models 4 and 5, are included in the table. Model 4 is a 13B dense model. Our 8×3.7B MoE architecture has fewer active parameters than this 13B model, leading to lower training and inference costs. Nevertheless, the 8×3.7B MoE model using Drop-Upcycling achieves better performance upon completion of training. Model 5 is a 3.7B dense model trained with 2.1T tokens. The fact that our 8×3.7B MoE model with Drop-Upcycling surpasses this dense model indicates that rather than continuously investing resources into training dense models, it might be a superior option to convert them to MoE models through Drop-Upcycling and continue training at a certain point in the process.

## 5.3 ANALYSIS 1: RE-INITIALIZAITON RATIO

We conducted a study to investigate the impact of the re-initialization ratio $r$ in Drop-Upcycling. Figure 4 illustrates the effects of different re-initialization rates 0.0 (naïve Upcycling), 0.1, 0.25, 0.5, 0.75, and 1.0 on models of sizes 8×152M and 8×1.5B. Each model was trained up to 150B tokens, during which we monitored the training loss and the progression of the average downstream task scores.

The experimental results revealed similar trends across both model sizes. In terms of long-term performance, a re-initialization ratio of 0.5 yielded the best results for both models, maintaining superiority in both training loss and average task scores. An interesting pattern emerged regarding the influence of the re-initialization ratio. With lower re-initialization rates, particularly at 0.0 (naïve Upcycling), the models struggled to significantly improve beyond the performance of the original pre-trained models. While re-initialization rates of 0.1 and 0.25 showed promising performance in the early stages of training, they were eventually surpassed by the 0.5 re-initialization rate as training progressed. These observations suggest that increasing the re-initialization ratio helps the models escape local optima, enabling more effective learning. However, excessively high re-initialization rates of 0.75 or 1.0 appeared to hinder the effective knowledge transfer from the pre-trained dense models. This phenomenon highlights an important trade-off concerning the MoE initialization: a balance must be struck between knowledge transfer and effective expert specialization. Drop-Upcycling with $r = 0.5$ is a robust and practical method that ideally balances these two aspects.

## 5.4 ANALYSIS 2: EXPERT SPECIALIZATION

We analyze expert routing patterns to examine how Drop-Upcycling facilitates expert specialization. We apply the methodologies of Jiang et al. (2024) and Muennighoff et al. (2024) to 8×1.5B MoE models trained with different methods. This analysis investigates how data from different domains is routed to various experts. As input data from different domains, we use the validation sets from

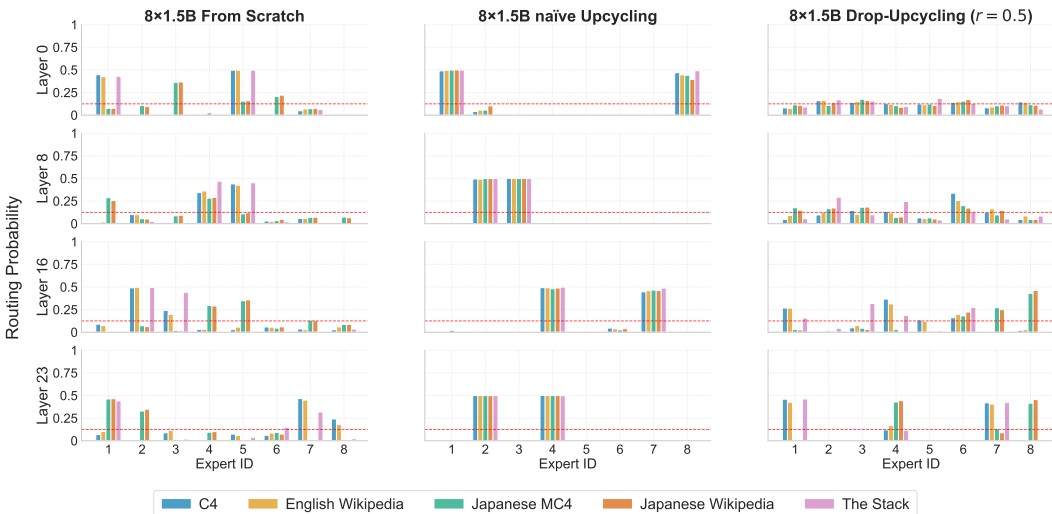

Figure 5: **Comparison of expert routing patterns across different MoE construction methods.** Drop-Upcycling exhibits more balanced expert utilization than naïve Upcycling. Results shown for layers 0 (first), 8, 16, and 23 (last); see Appendix C.2 for results on all layers.

Japanese and English Wikipedia; the validation set of the Japanese MC4 dataset (as split by the authors; see LLM-jp 2024), originally introduced by Raffel et al. (2019); The Stack (Kocetkov et al., 2023); and the English C4 dataset (Muennighoff et al., 2024).

In Figure 5, we observe that naïve Upcycling with global load balancing results in a highly imbalanced routing pattern, where the majority of experts were underutilized or not utilized at all, with only two experts being always selected across all layers. While layer-wise load balancing mitigate such expert collapse, we found no significant differences in the training loss trajectories or model performance between these two strategies (see Appendix C.3). In contrast, both the model trained from scratch and the one enhanced with Drop-Upcycling (with $r = 0.5$) exhibit domain-specialized routing patterns regardless of the load balancing strategy. The routing patterns reveal that certain experts specialize in processing specific types of data, such as Japanese text, English text, or code snippets, as evident from the distinct expert selection probabilities corresponding to each dataset.

These findings suggest that Drop-Upcycling promotes effective expert specialization independently of the load balancing strategy, which likely contributes to the improved performance observed in our experiments. For detailed routing patterns across all 24 layers and further analysis of load balancing strategies, see Appendix C.2 and C.3.

## 6 CONCLUSION

In this paper, we introduced Drop-Upcycling, a novel method for efficiently constructing Mixture of Experts (MoE) models from pre-trained dense models. Selectively re-initializing parameters of expert feedforward networks, Drop-Upcycling effectively balances knowledge transfer and expert specialization, addressing the key challenges in MoE model development.

Our extensive large-scale experiments demonstrated that Drop-Upcycling, significantly outperforms previous MoE construction methods. As a result, we achieved an MoE model with 5.9B active parameters that matches the performance of a 13B dense model from the same model family while requiring only about 1/4 of the training FLOPs.

By making all aspects of our research publicly available—including data, code, configurations, checkpoints, and logs—we aim to promote transparency and facilitate further advancements in efficient LLM training. We believe that Drop-Upcyclingoffers a practical solution to reduce resource barriers in deploying high-performance LLMs, contributing to broader accessibility and innovation in AI research.

ACKNOWLEDGEMENTS

The authors would like to thank Masanori Suganuma and Kou Misaki for providing valuable discussions and feedback during the preparation of this manuscript. This work was supported by the "R&D Hub Aimed at Ensuring Transparency and Reliability of Generative AI Models" project of the Ministry of Education, Culture, Sports, Science and Technology. This study was carried out using the TSUBAME4.0 supercomputer at Institute of Science Tokyo.

AUTHOR CONTRIBUTIONS

Taishi Nakamura initiated the project, designed the method, and carried out the experiments. Takuya Akiba co-designed the experiments and formulated the overall research strategy. Kazuki Fujii implemented the training codebase used for the experiments. Yusuke Oda handled the training of the dense models. Jun Suzuki and Rio Yokota provided guidance and oversight throughout the project. All authors contributed to the writing and approved the final manuscript.

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

Table 3: Detailed FLOPs Breakdown for Transformer Models (Single Forward Pass)

| Component | FLOPs |
|---|---|
| Embeddings | $2svd_h$ |
| Attention (per layer) | |
|    Key and value projections | $4sd_hd_kn_q$ |
|    Query projections | $2sd_hd_kn_h$ |
|    Key @ Query logits | $2s^2d_kn_h$ |
|    Attention matrix computation | $2s^2d_kn_h$ |
|    Softmax @ value reductions | $2sd_kn_hd_h$ |
| FFN (SwiGLU, per layer) | |
|    Dense model | $4sd_hd_f + 2sd_fd_h$ |
|    MoE model | $n_e(4sd_hd_f + 2sd_fd_h)$ |
| Final Logits | $2sd_hv$ |
| Total (Dense) | embeddings $+n_l(attention + \text{ffn}_{\text{Dense}})+$ logits |
| Total (MoE) | embeddings $+n_l(attention + \text{ffn}_{\text{MoE}})+$ logits |

# A    EXPERIMENTAL SETUP DETAILS

## A.1    FLOPS CALCULATION

Table 3 presents the method for calculating FLOPs (floating point operations) for the forward path in transformer components. The variables used are as follows: $s$ (sequence length), $d_h$ (hidden size), $v$ (vocabulary size), $d_f$ (FFN intermediate size), $n_l$ (number of layers), $n_h$ (number of attention heads), $n_q$ (number of query groups), $d_k$ (attention head dimension), and $n_e$ (number of selected experts per token). For matrix multiplication $A_{m\times k} \times X_{k\times n}$, $2m \times k \times n$ FLOPs are required in the forward pass (the factor of 2 accounts for both multiplication and addition operations). The table displays the main FLOPs contributors for the forward path only. It should be noted that the computational costs for sigmoid and Hadamard product within SwiGLU calculations, MoE gate computations, and RMS Norm calculations are considered negligible and thus omitted from this analysis. While not shown in the table, backward propagation typically requires approximately twice the FLOPs of forward propagation.

## A.2    IMPLEMENTATION AND TRAINING ENVIRONMENT

For our experiments with MoE models and the training of the 1.5B Dense model, we utilized the TSUBAME 4.0 supercomputer at the Global Scientific Information and Computing Center, Institute of Science Tokyo. This environment is equipped with NVIDIA H100 SXM5 94GB GPUs, with each node housing 4 H100 GPUs. Inter-node communication is facilitated by InfiniBand NDR200 interconnects. The training of our largest model, the 8×3.7B model, employed 16 nodes (totaling 64 GPUs). For the training of the 152M and 3.7B Dense models, we leveraged the high-performance computing nodes (PHY) provided by Sakura Internet. This setup features NVIDIA H100 80GB GPUs, with each node containing 8 H100 GPUs. The network interface is equipped with four 400Gb RoCEv2-compatible NICs and two 25Gb NICs. The training of our largest Dense model (3.7B parameters) utilized a maximum of 32 nodes (totaling 256 GPUs).

For implementation, we used Megatron-LM[2] for Dense model training, and moe-recipes[3] for MoE model training. Additionally, Flash Attention 2 (Dao, 2024) was utilized to improve computational efficiency and reduce memory usage. All the training processes were conducted using bfloat16 precision.

Table 4: Model Configuration Details

| Model | Act Params / Total Params | Layers | $d_{model}$ | $d_{ff}$ | Attn Heads | KV Heads | Vocab Size |
|---|---|---|---|---|---|---|---|
| Dense 152M | 152M / 152M | 12 | 512 | 2,048 | 8 | 8 | 99,574 |
| Dense 1.5B | 1.5B / 1.5B | 24 | 2,048 | 7,168 | 16 | 8 | 48,586 |
| Dense 3.7B | 3.7B / 3.7B | 28 | 3,072 | 8,192 | 24 | 24 | 99,574 |
| Dense 13B | 13B / 13B | 40 | 5,120 | 13,824 | 40 | 40 | 99,574 |
| MoE 8×152M | 190M / 417M | 12 | 512 | 2,048 | 8 | 8 | 99,574 |
| MoE 8×1.5B | 2.6B / 8.9B | 24 | 2,048 | 7,168 | 16 | 8 | 48,586 |
| MoE 8×3.7B | 5.9B / 18B | 28 | 3,072 | 8,192 | 24 | 24 | 99,574 |

### A.3 MODEL CONFIGURATIONS

As described in Section 4, we selected the Llama (Touvron et al., 2023) and Mixtral (Jiang et al., 2024) architectures for dense and MoE models, respectively, for our experiments. Both architectures are based on the Transformer (Vaswani et al., 2017) with several improvements, including RMSNorm (Zhang & Sennrich, 2019), SwiGLU (Shazeer, 2020), and rotary position embeddings (RoPE) (Su et al., 2024). The notable difference in Mixtral (MoE) from Llama (dense) is that all feedforward network (FFN) layers are replaced by sparsely gated MoE layers. Table 4 shows the details of the model configuration.

### A.4 MODEL TRAINING CONFIGURATIONS

As shared settings for training all models, we adopted the following hyperparameters: AdamW optimizer (Loshchilov & Hutter, 2019) with $\beta_1 = 0.9$, $\beta_2 = 0.95$, and $\epsilon = 10^{-8}$, sequence length of 4096, weight decay of 0.1, and gradient clipping of 1.0. The global batch size was set to 1024 for the 1.5B, 3.7B and 13B models, and 512 for the 152M model.

We used cosine decay for learning rate scheduling. For Dense models, the maximum learning rate was set to $3 \times 10^{-4}$, and it decayed to $3 \times 10^{-5}$ over 1,000B tokens for the 1.5B model, and 2,072B tokens for the 152M, 3.7B and 13B models, with the learning rate remaining constant during the final 2000 steps. For MoE models, the maximum learning rate was set to $2 \times 10^{-4}$, and it decayed to $2 \times 10^{-5}$ over 500B tokens. Additionally, to prevent instability in training due to unbalanced routing on the MoE models, a load balancing loss was introduced, with the coefficient unified at 0.02 across all MoE models.

## B DATASETS AND EVALUATION METHODS

### B.1 TRAINING DATASET DETAILS

We used the LLM-jp corpus v3[4], an open corpus curated by the LLM-jp working group, for training English and Japanese bilingual language models. The corpus consists of 1.7T tokens in English, Japanese, and source code with a small amount of Chinese and Korean tokens. Following the LLM-jp's scheme, some Japanese portion of the corpus is upsampled by 2 to obtain 2.1T training tokens in total.

Table 5 describes the statistics of the corpus subsets that were used for training data of the Dense and MoE models in our experiments.

Table 6 details the dataset distribution percentages used for training the different model sizes. The 152M, 3.7B, and 13B models share the same data proportions, while the 1.5B model has slightly different percentages.

---

[2]`https://github.com/NVIDIA/Megatron-LM`

[3]`https://github.com/rioyokotalab/moe-recipes`, Version 1.0.0

[4]https://gitlab.llm-jp.nii.ac.jp/datasets/llm-jp-corpus-v3

Table 5: Statistics of the training dataset.

| Language | Subset | #tokens [$\times 10^9$] |
|---|---|---|
| English | Dolma 1.6 (sampled) (Soldaini et al., 2024) | 945. |
| | Wikipedia | 4.7 |
| Japanese | Common Crawl (LLM-jp, 2024) | 381. |
| | Kaken | 0.9 |
| | NDL WARP HTML | 1.3 |
| | NDL WARP PDF | 207. |
| | Wikipedia | 1.3 |
| Chinese | Wikipedia | 0.8 |
| Korean | Wikipedia | 0.9 |
| Code | The Stack (Kocetkov et al., 2023) | 114. |

Table 6: Dataset Distribution Overview (Percentages)

| Language | Subset | 152M/3.7B/13B | 1.5B |
|---|---|---|---|
| English | Dolma | 45.6% | 39.7% |
| | Wikipedia | 0.2% | 0.5% |
| Japanese | Common Crawl | 36.8% | 49.5% |
| | Kaken | 0.1% | 0.1% |
| | NDL WARP HTML | 0.1% | - |
| | NDL WARP PDF | 11.5% | - |
| | Wikipedia | 0.1% | 0.2% |
| Chinese | Wikipedia | 0.1% | - |
| Korean | Wikipedia | 0.1% | - |
| Code | The Stack | 5.5% | 10.1% |
| Total Tokens (B) | | 2,072 | 1,000 |

## B.2 EVALUATION DATASETS AND METHODOLOGIES

Table 7 provides detailed information about the evaluations used in our experiments. The evaluation tasks comprise both Japanese and English language assessments. We utilized publicly available evaluation code for our assessments[5].

The evaluation tasks are categorized into seven types, such as free-form QA (NIILC (Sekine, 2003), JEMHQA (Ishii et al., 2023)), machine reading comprehension (JSQuAD (Kurihara et al., 2022), SQuAD2 (Rajpurkar et al., 2018)), abstractive summarization (XL-Sum (Hasan et al., 2021)), machine translation (WMT'20 En-Ja, Ja-En (Barrault et al., 2020)), question answering (Open-BookQA (Mihaylov et al., 2018), TriviaQA (Joshi et al., 2017)), common sense reasoning (HellaSwag (Zellers et al., 2019), XWinograd (Tikhonov & Ryabinin, 2021)), and logical reasoning (Big Bench Hard (BBH) (Suzgun et al., 2023)). We used 4-shot prompting for the Free-form QA, machine reading comprehension, machine translation, question answering, and commonsense reasoning tasks, 1-shot prompting for the abstractive summarization task, and 3-shot prompting for the logical reasoning task. Moreover, we also applied the Chain-of-Thought method (Wei et al., 2022) for the logical reasoning task.

---

[5]`https://github.com/swallow-llm/swallow-evaluation`

Table 7: Evaluation Benchmark Details

| | JEM HQA | NIILC | JSQ | XL-Sum | WMT E→J | WMT J→E | OB QA | TQA | HS | SQ v2 | XW-EN | BBH |
|---|---|---|---|---|---|---|---|---|---|---|---|---|
| Dataset | JEMHQA | NIILC | JSQuAD | XL-Sum | WMT20 | | OBQA | TriviaQA | HellaSwag | SQuAD2 | XWINO | BBH |
| Task | | QA | MRC | Summ. | Trans. | | | QA | MRC | MRC | Commonsense Reasoning | Logical Reasoning |
| Language | JA | JA | JA | JA | EN→JA | JA→EN | EN | EN | EN | EN | EN | EN |
| # Instances | 120 | 198 | 4,442 | 766 | 1,000 | 993 | 500 | 17,944 | 10,042 | 11,873 | 2,325 | 6,511 |
| Few-shot # | 4 | 4 | 4 | 1 | 4 | 4 | 4 | 4 | 4 | 4 | 4 | 3 |
| Evaluation Metric | | Character F1 | | ROUGE-2 | | BLEU | | | Accuracy | | | CoT Acc. |

Table 8: Gate Initialization Pattern Comparison for 8×1.5B Models (Training Tokens: 50B)

| | Initialization | Results | | | | | | | | | | | | |
|---|---|---|---|---|---|---|---|---|---|---|---|---|---|---|
| # | Distribution | JEM | NII | JSQ | XL | J→E | E→J | OBQ | TrQ | SQ2 | HeS | XWI | BBH | AVG |
| 1 | $\mathcal{N}(0, 0.02)$ | 46.1 | 37.9 | **63.6** | 9.2 | 15.4 | 8.1 | 22.4 | **19.4** | **41.7** | **15.6** | 80.0 | 25.9 | 32.1 |
| 2 | $\mathcal{N}(0, 0.2887)^*$ | 50.6 | 38.6 | 54.6 | 9.3 | 15.5 | **8.3** | 20.6 | 18.4 | 41.1 | 14.3 | 79.8 | 24.7 | 31.3 |
| 3 | $\mathcal{U}(-0.0346, 0.0346)^\dagger$ | 49.2 | **38.9** | 61.0 | **9.7** | **16.0** | 7.9 | **23.6** | 18.9 | **41.7** | 15.5 | **80.9** | 23.9 | **32.3** |
| 4 | $\mathcal{U}(-0.5, 0.5)$ | 44.6 | 36.3 | 56.3 | 8.6 | 15.5 | 8.1 | 20.6 | 17.7 | 41.0 | 14.6 | 80.0 | **26.0** | 30.8 |
| 5 | $\mathcal{U}(0, 1)$ | **51.5** | 36.8 | 55.6 | 9.0 | 15.7 | 7.9 | 21.6 | 18.3 | 41.0 | 15.3 | 80.1 | 25.1 | 31.5 |

$\mathcal{N}(\mu, \sigma)$: Normal distribution with mean $\mu$ and standard deviation $\sigma$.
$\mathcal{U}(a, b)$: Uniform distribution over the interval $[a, b]$.
$^*$ $\sigma = \sqrt{1/12} \approx 0.2887$, matches the standard deviation of $\mathcal{U}(0, 1)$.
$^\dagger$ Corrected from $\mathcal{U}(-0.0346, 0.0346)$ to match the standard deviation of 0.02. Bold values indicate the best score for each task.

## C   ADDITIONAL EXPERIMENTAL RESULTS AND ANALYSIS

### C.1   COMPARISON OF GATE INITIALIZATION METHODS

We conducted a detailed investigation into the effects of gate initialization on the performance of naïve Upcycling. An ablation study was performed on five different initialization patterns. Table 8 presents the comparison results of different gate initialization patterns in an 8×1.5B model. Performance was evaluated after training on 50B tokens.

While preliminary experiments had indicated better results with a standard deviation of 0.28, our main experiments revealed that a uniform distribution with a standard deviation of 0.02 achieved the highest average performance across tasks. Based on these results, we adopted a uniform distribution ($\mathcal{U}(-0.0346, 0.0346)$, as the standard method for gate initialization in this study. It is worth noting that gate initialization may not be a critical factor in model performance, and any initialization that avoids extreme values such as excessively high standard deviations is likely to be sufficient.

### C.2   DETAILED ANALYSIS OF EXPERT ROUTING PATTERNS ACROSS LAYERS

For a comprehensive view of routing patterns across all layers, we provide detailed plots of expert routing probabilities for all 24 layers, grouped into early, middle, and late stages. These plots offer a more granular analysis of how routing behaviors evolve throughout the model depth.

Figures 6 to 8 show the expert routing patterns for all 24 layers of the 8×1.5B MoE models trained with different methods, grouped into early (layers 0-7), middle (layers 8-15), and late (layers 16-23) stages. This comprehensive view allows for a detailed analysis of how routing patterns evolve across the entire model depth.

These figures illustrate how the routing patterns evolve throughout the model layers, providing insights into the specialization and behavior of experts at different depths. Notably, the naïve Upcycling method does not exhibit clear evidence of bias towards specific domains in any layer. In contrast, our proposed method demonstrates domain specialization in multiple layers across the network—from those closest to the input to those near the output—while reusing the parameters of the dense model. This indicates that our approach effectively facilitates expert specialization in several

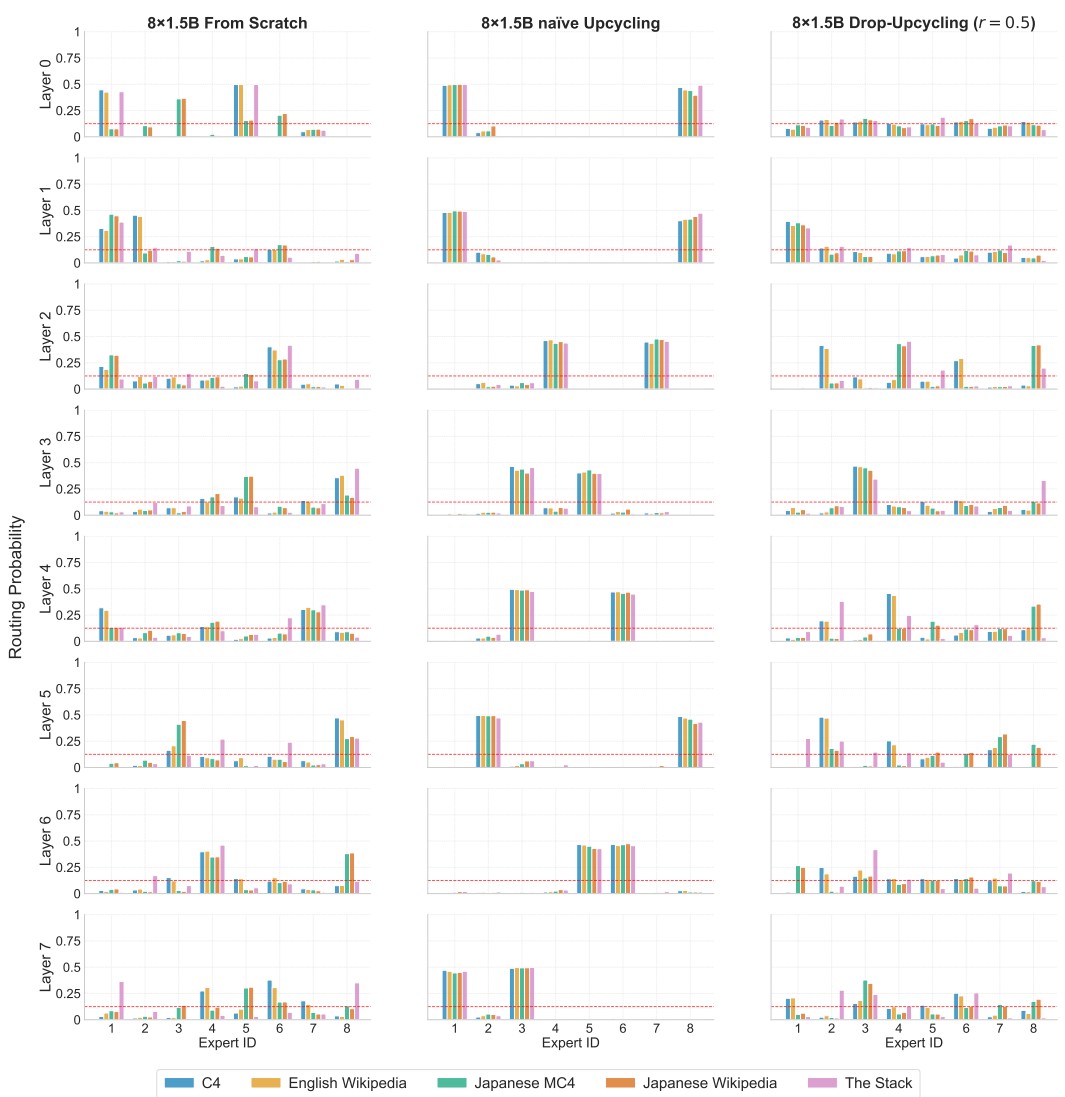

Figure 6: Expert routing patterns for early layers (0-7) of the 8×1.5B MoE models.

layers without the need to train from scratch, leveraging the pre-trained dense model to achieve efficient domain-specific routing throughout significant portions of the network depth.

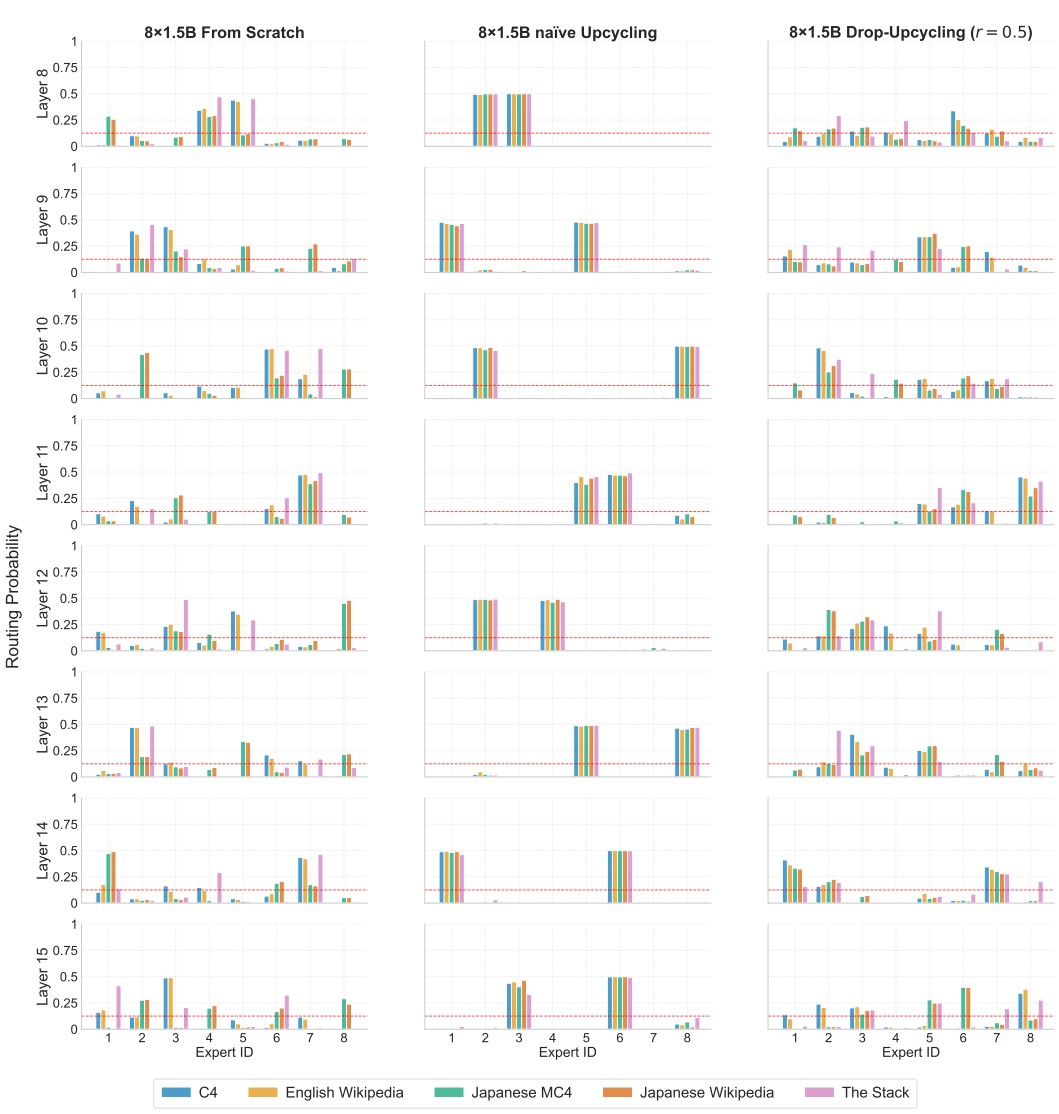

Figure 7: Expert routing patterns for middle layers (8-15) of the 8×1.5B MoE models.

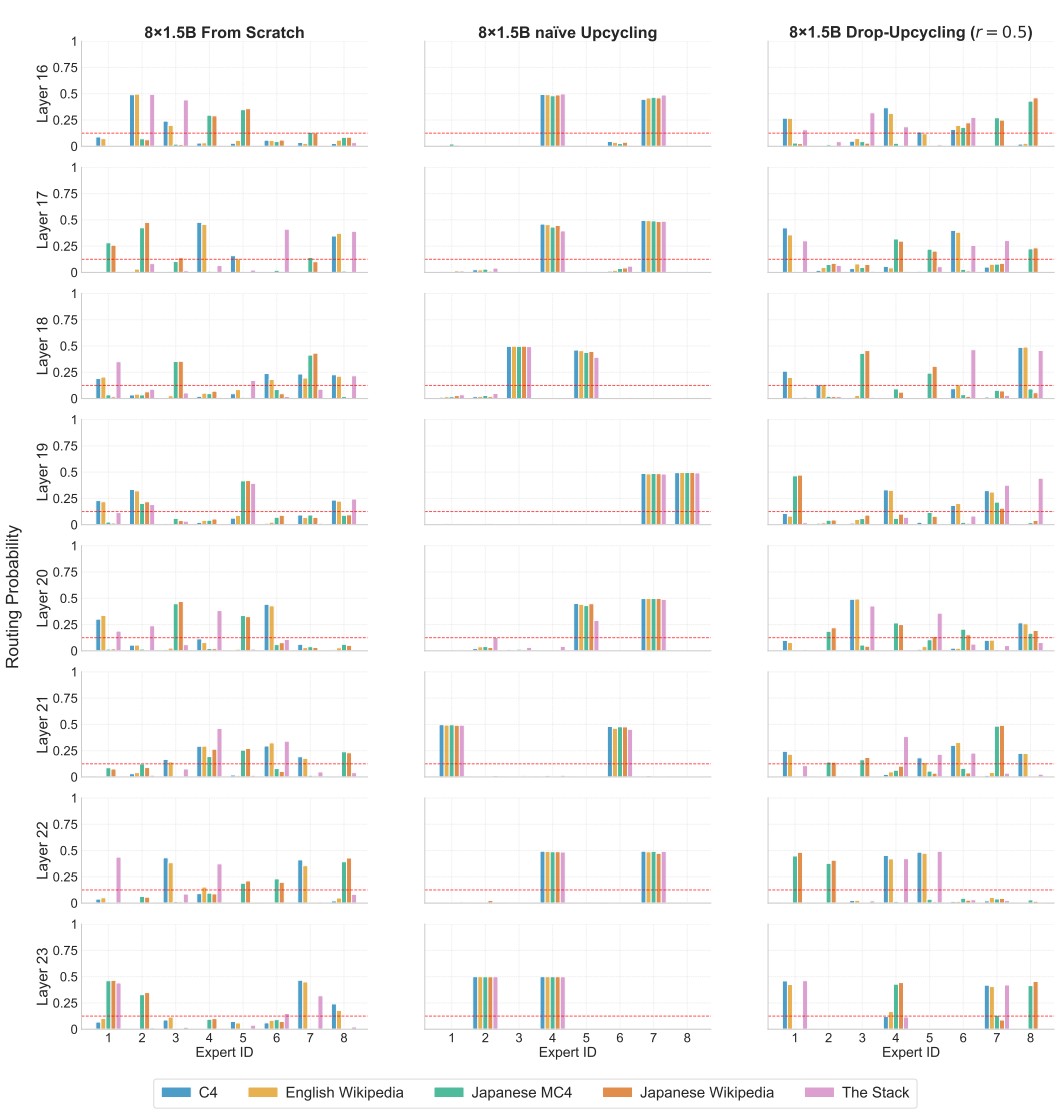

Figure 8: Expert routing patterns for late layers (16-23) of the 8×1.5B MoE models.

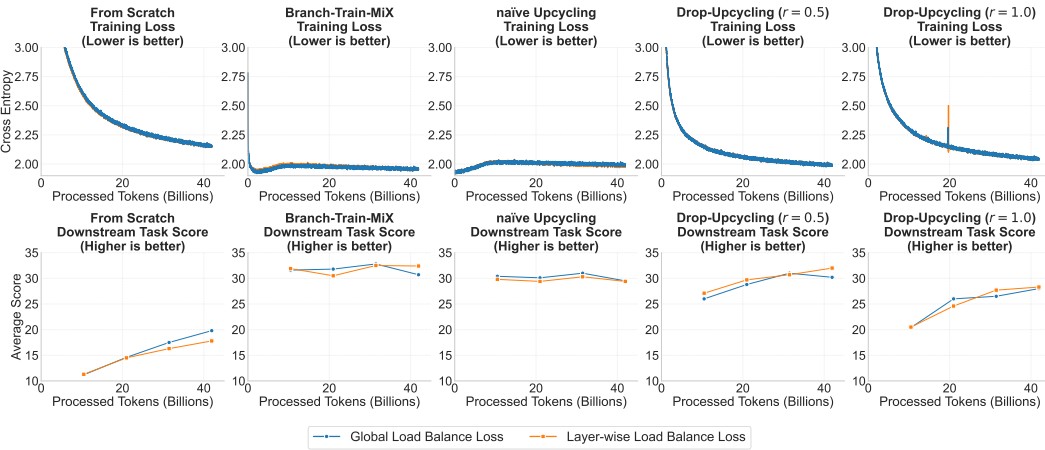

Figure 9: **Comparison between global and layer-wise load balancing across different initialization methods.** Top: Training loss trajectories over 40B tokens. Bottom: Evaluation metrics measured at iterations corresponding to 10B, 20B, 30B, and 40B tokens. Results show comparable performance between global and layer-wise approaches across all methods.

### C.3 COMPARING GLOBAL VS. LAYER-WISE LOAD BALANCING

In our experiments (Section 5), we applied load balancing loss globally rather than layer-wise. This approach aligns with the implementation in the HuggingFace Transformers library and is widely adopted in the community. To analyze the effect of global and layer-wise load balancing, we conducted a comparative analysis between global and layer-wise load balancing applications across 40B tokens for different initialization methods (From Scratch, Branch-Train-MiX, naïve Upcycling, and Drop-Upcycling with r=0.5 and r=1.0) in the 8×1.5B setting. As shown in Figure 9, both approaches yield similar training loss trajectories and downstream task performance. These results suggest that the effectiveness of Drop-Upcycling is not significantly affected by whether load balancing loss is applied globally or layer-wise.

Figures 10 through 12 show the routing patterns when applying layer-wise load balancing loss at 40B tokens. The results demonstrate that Drop-Upcycling (r=0.5) exhibits domain-specialized routing patterns similar to training from scratch. In contrast, naïve Upcycling shows nearly uniform routing across all layers except the final layer, which aligns with findings reported in Jiang et al. (2024). Our proposed Drop-Upcycling method appears to escape the local optima observed in naïve Upcycling, which likely contributes to its improved performance.

The trade-offs between layer-wise and global load balancing—whether to enforce uniform expert utilization through layer-wise application or to allow potential expert collapse with global application—along with broader questions about MoE architecture design (such as varying expert counts per layer) remain as interesting directions for future research.

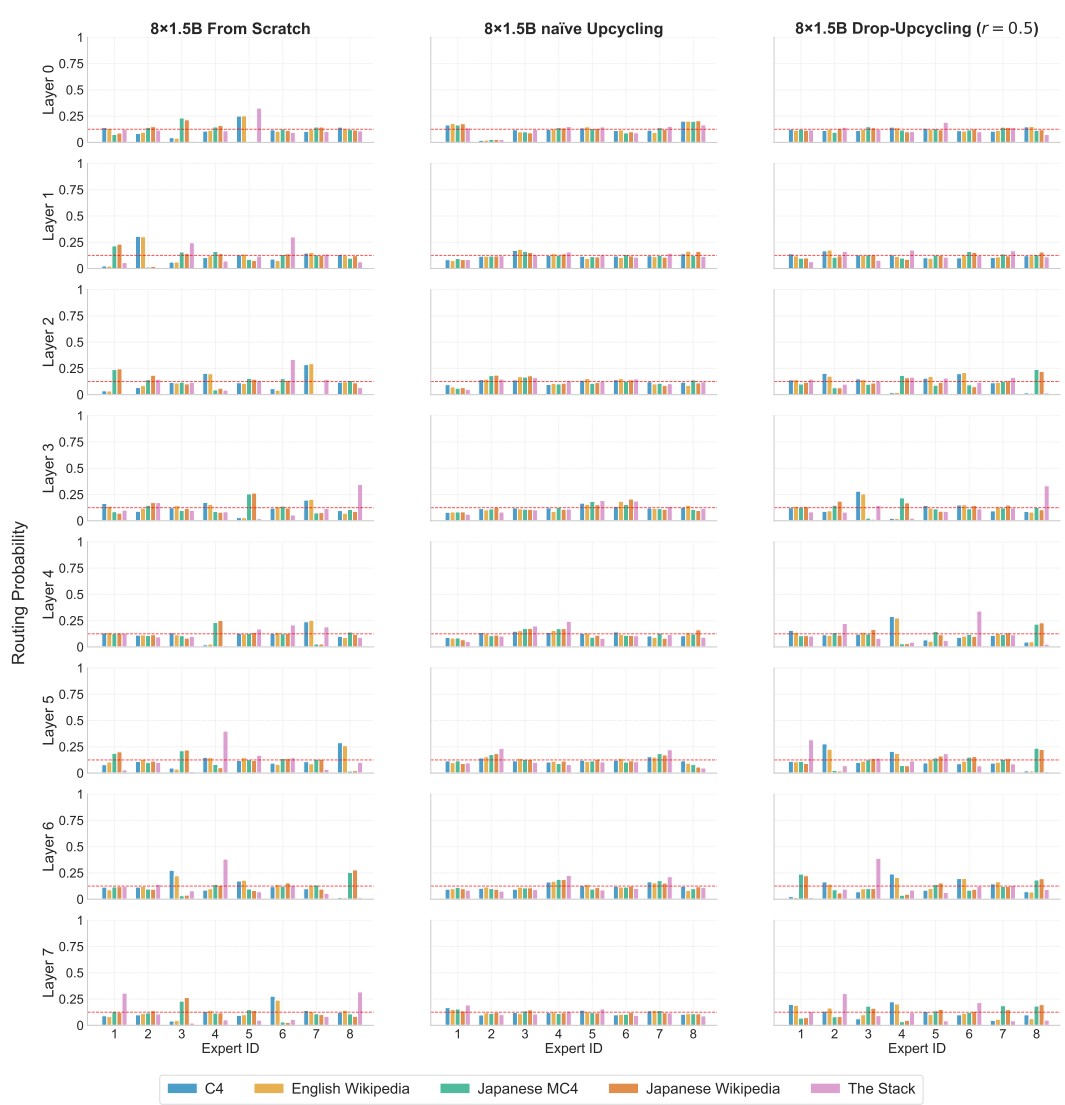

Figure 10: Expert routing patterns for early layers (0-7) under layer-wise load balancing at 40B tokens

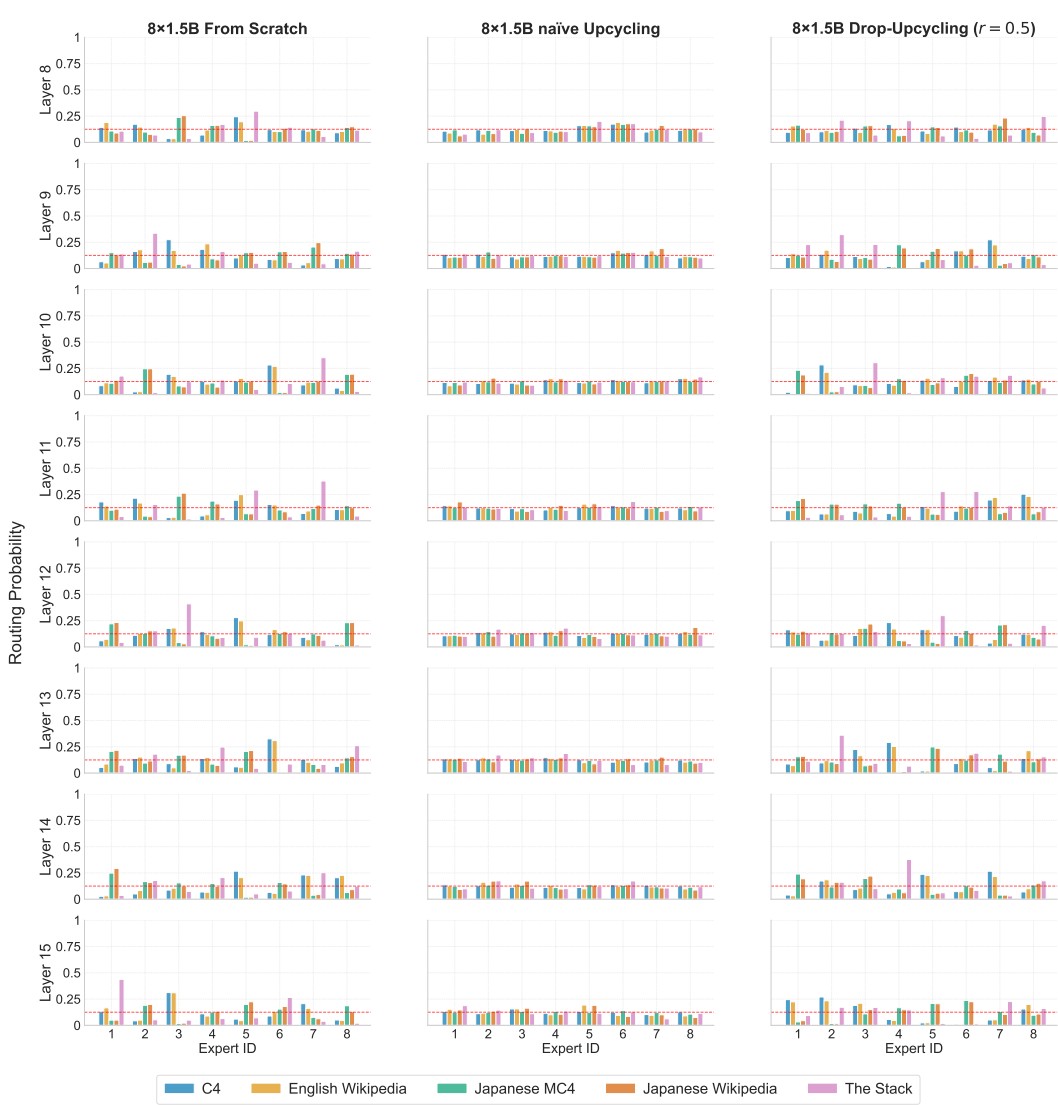

Figure 11: Expert routing patterns for middle layers (8-15) under layer-wise load balancing at 40B tokens

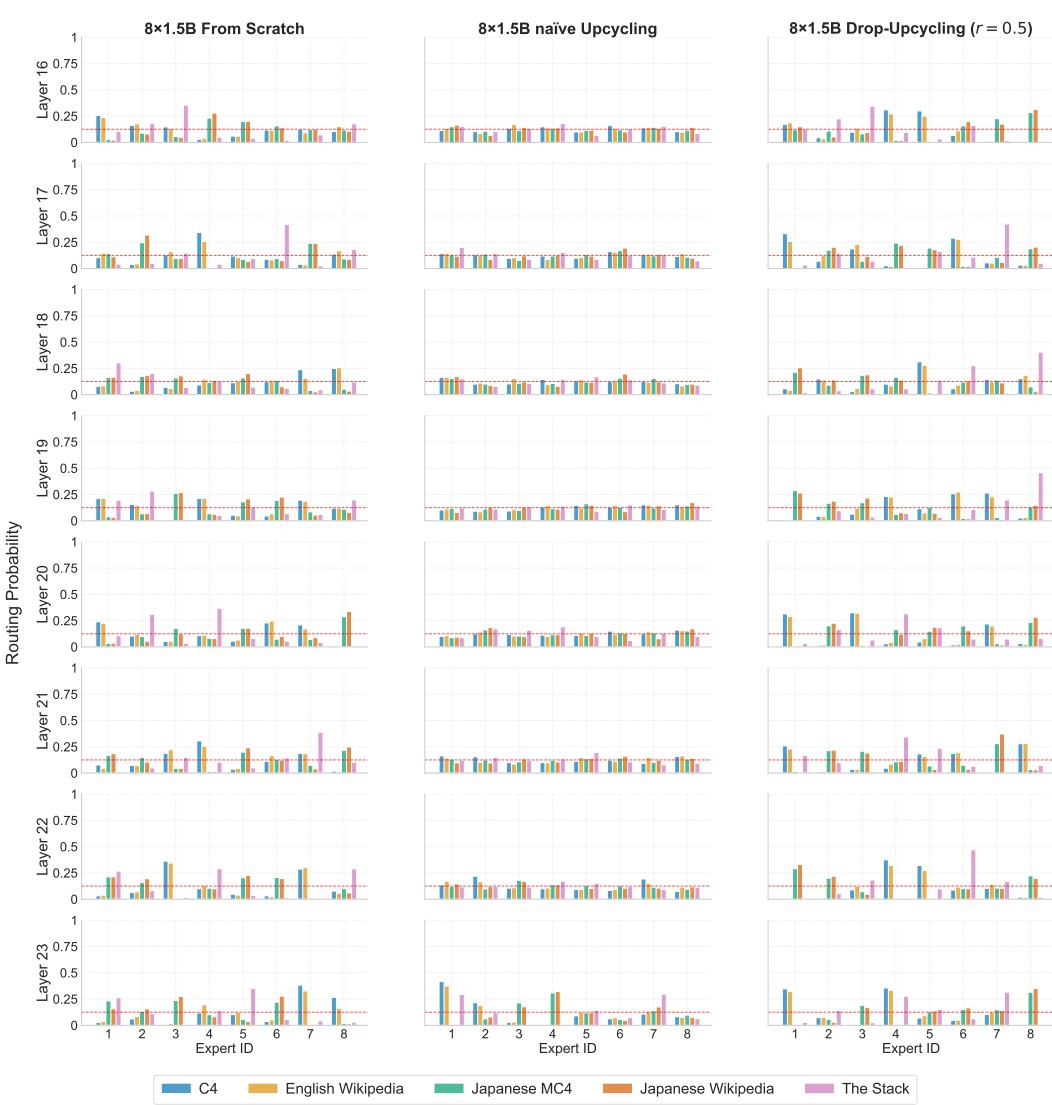

Figure 12: Expert routing patterns for late layers (16-23) under layer-wise load balancing at 40B tokens

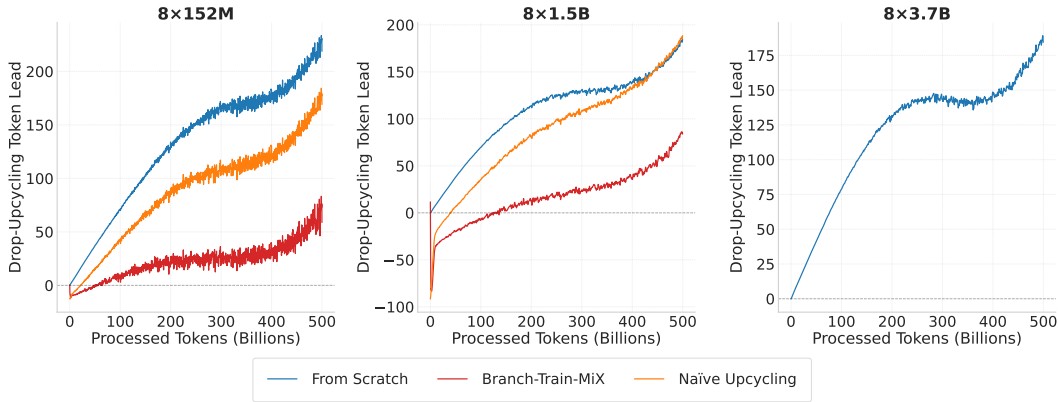

Figure 13: **Convergence catch-up analysis.** We compare the relative convergence speed of Drop-Upcycling and baseline methods by examining the number of training tokens required to reach the same loss value. The x-axis represents the number of training tokens processed by the baseline method, while the y-axis shows the difference in training tokens needed by Drop-Upcycling to achieve the same loss. Positive values indicate that Drop-Upcycling achieves the loss faster, while negative values suggest the baseline method is ahead.

### C.4 CONVERGENCE CATCH-UP ANALYSIS

To examine the selection of methods based on the training budget and to explore potential extrapolations of long-term trends beyond the scope of our analysis so far, we conduct a brief relative quantitative analysis of the convergence speeds of Drop-Upcycling and baseline methods. In Figure 13, we compare the number of training tokens required to reach the same loss value for Drop-Upcycling and the baseline methods. The plot shows that no significant trend of diminishing advantage for Drop-Upcycling over the baseline methods is observed. This indicates that training from scratch would require an impractically large number of tokens to match Drop-Upcycling, making Drop-Upcycling the better choice in practical scenarios.

However, it is important to acknowledge the limitations of this analysis. First, the effect of the learning rate (LR) schedule must be considered. Differences in LR due to different step counts could artificially influence the observed trends in convergence advantage. For example, we hypothesize that the widening advantage of Drop-Upcycling observed late in training (after 400B tokens) may not entirely reflect the contribution of Drop-Upcycling itself but could instead be attributed to the influence of LR scheduling. To eliminate the impact of LR scheduling, conducting all experiments with a constant LR would provide a more valid basis for this comparison.

Second, it is worth noting that Branch-Train-Mix utilizes an additional training budget for pretraining individual experts before MoE training. In our setup, for instance, three expert models were pretrained using 100B tokens each, requiring a total of 300B tokens for dense model training before the MoE training phase. As a result, while Branch-Train-Mix appears to show an initial advantage in the plot, this advantage diminishes when accounting for the total training budget. Thus, in terms of overall efficiency, Branch-Train-Mix offers little to no advantage during most of the training process.

### C.5 DETAILED DERIVATIONS OF THEORETICAL CHARACTERISTICS

Consider the output of MoE layer with parameter re-initialization ratio $r$. Let $\text{FFN}_{\text{retained}_i}(\mathbf{x})$ denote the output from expert $i$'s preserved original parameters (ratio $(1-r)$) and $\text{FFN}_{\text{diverse}_i}(\mathbf{x})$ denote the output from reinitialized parameters (ratio $r$). The exact form of MoE output is:

$$\mathbf{y} = \sum_{i=1}^{N} g(\mathbf{x})_i \cdot \left( \text{FFN}_{\text{retained}_i}(\mathbf{x}) + \text{FFN}_{\text{diverse}_i}(\mathbf{x}) \right) \qquad (6)$$

where $g(\mathbf{x})_i$ is the gating function defined in 3. Note that $g(\mathbf{x})_i = 0$ for experts not among the top-$k$ selected.

Let $S_k$ denote the set of indices for the $k$ selected experts. We can rewrite the output as:

$$
\begin{aligned}
\mathbf{y} &= \sum_{i \in S_k} g(\mathbf{x})_i \cdot (\text{FFN}_{\text{retained}_i}(\mathbf{x}) + \text{FFN}_{\text{diverse}_i}(\mathbf{x})) \\
&= \sum_{i \in S_k} g(\mathbf{x})_i \cdot [\text{FFN}_{\text{common}}(\mathbf{x}) + (\text{FFN}_{\text{retained}_i}(\mathbf{x}) - \text{FFN}_{\text{common}}(\mathbf{x})) + \text{FFN}_{\text{diverse}_i}(\mathbf{x})] \\
&= \text{FFN}_{\text{common}}(\mathbf{x}) \sum_{i \in S_k} g(\mathbf{x})_i + \sum_{i \in S_k} g(\mathbf{x})_i \cdot [\text{FFN}_{\text{retained}_i}(\mathbf{x}) - \text{FFN}_{\text{common}}(\mathbf{x}) + \text{FFN}_{\text{diverse}_i}(\mathbf{x})] \\
&= \text{FFN}_{\text{common}}(\mathbf{x}) + \sum_{i \in S_k} g(\mathbf{x})_i \cdot [\text{FFN}_{\text{retained}_i}(\mathbf{x}) - \text{FFN}_{\text{common}}(\mathbf{x}) + \text{FFN}_{\text{diverse}_i}(\mathbf{x})] \\
&= \text{FFN}_{\text{common}}(\mathbf{x}) + \sum_{i=1}^{N} g(\mathbf{x})_i \cdot [\text{FFN}_{\text{retained}_i}(\mathbf{x}) - \text{FFN}_{\text{common}}(\mathbf{x}) + \text{FFN}_{\text{diverse}_i}(\mathbf{x})]
\end{aligned}
\tag{7}
$$

where the third equality follows from distributing the sum, the fourth equality follows from $\sum_{i \in S_k} g(\mathbf{x})_i = 1$, and the final equality holds because $g(\mathbf{x})_i = 0$ for $i \notin S_k$. Here $\text{FFN}_{\text{common}}(\mathbf{x})$ represents the output from parameters common to all selected experts.

For each expert, a ratio $(1 - r)$ of parameters are randomly preserved from the original FFN. When $k$ experts are selected, the probability that a parameter is preserved in all $k$ experts is $(1 - r)^k$. Therefore, approximately $(1 - r)^k \cdot d_f$ dimensions have common preserved parameters among selected experts, where $d_f$ is the intermediate dimension size. Note that beyond these completely common parameters, there may be partial parameter sharing among subsets of the selected experts due to the random preservation process.

To understand the error bound $O(\frac{1}{\sqrt{d_f}})$, consider that for any two experts $i, j$, the number of overlapping parameters follows a binomial distribution $B(d_f, (1 - r)^2)$. By the Central Limit Theorem, the deviation from the expected value scales with $\sqrt{d_f}$, leading to a relative error of $O(\frac{1}{\sqrt{d_f}})$ in the parameter overlap estimation.

### C.6 Extensions to Fine-grained and Shared Experts

We discuss the natural extension of Drop-Upcycling to advanced MoE architectures: fine-grained experts and shared experts proposed in DeepSeekMoE (Dai et al., 2024). For an original dense FFN with hidden dimension $d_h$ and intermediate size $d_f$, DeepSeekMoE introduces granularity parameter $m$ to split each of $N$ experts into finer segments (each with intermediate size $d_f/m$), where $mk$ experts are selected by top-$mk$ routing, and $k_s$ shared experts process all tokens. The total number of experts becomes $mN$ with $mk$ nonzero gates, which reduces to $mN - k_s$ experts and $mk - k_s$ gates when using shared experts.

#### C.6.1 Extension to Fine-grained MoE

For simplicity of discussion, we assume $d_f$ is divisible by $m$ for fine-grained MoE (a realistic assumption since $m$ is typically a power of 2 and $d_f$ contains powers of 2 as factors). The output of the MoE layer is expressed as:

$$
y = \sum_{i=1}^{mN} g(x)_{(i)} \cdot \text{FFN}_{(i)}(x)
\tag{8}
$$

When applying Drop-Upcycling to convert from a dense FFN layer to a fine-grained MoE layer, we conduct the following steps:

1. Expert Dimension Sampling. First, randomly sample $d_f/m$ dimensions from the original FFN intermediate dimension $d_f$ for each expert.
2. Column-wise Reinitialization Sampling. For each expert's sampled $d_f/m$ dimensions, select an index set $\mathcal{S}$ where $|\mathcal{S}| = \lfloor r \cdot d_f/m \rfloor$ dimensions to be reinitialized.
3. Statistics Calculation. Calculate means and standard deviations $(\mu_{\text{up}}, \sigma_{\text{up}})$, $(\mu_{\text{gate}}, \sigma_{\text{gate}})$, $(\mu_{\text{down}}, \sigma_{\text{down}})$ for the weight matrices corresponding to the selected indices $\mathcal{S}$.

4. Partial Re-Initialization. Initialize each expert's weight matrices according to:

$$\widetilde{\mathbf{W}}_{\text{type}} = \mathbf{I}_{\mathcal{S}} \odot \mathbf{R}_{\text{type}} + (1 - \mathbf{I}_{\mathcal{S}}) \odot \mathbf{W}_{\text{type}} \tag{9}$$

where $\mathbf{R}_{\text{type}}$ is sampled from $\mathcal{N}(\mu_{\text{type}}, (\sigma_{\text{type}})^2)$.

Note that the portion reinitialized by our method needs to be scaled down due to the increased number of activated experts in top-$mk$ routing resulting in smaller $g(\mathbf{x})_i$. While the absolute magnitude information in router outputs might adapt during training, following He et al. (2024), scaling the weights of $W_{\text{down}}$ and $W_{\text{up}}$ might be beneficial.

### C.6.2 COMBINATION WITH SHARED EXPERTS

When using both shared experts and fine-grained experts, the output is:

$$y = \sum_{i=1}^{k_s} \text{FFN}_{(i)}(x) + \sum_{i=k_s+1}^{mN} g(x)_{(i-k_s)} \cdot \text{FFN}_{(i)}(x) \tag{10}$$

Here, shared experts are always active and process dimensions $(d_h, d_f/m \cdot k_s)$, while fine-grained experts each process $d_f/m$ dimensions.

We initialize fine-grained experts using the method described above. For shared experts, we can either randomly sample $d_f/m \cdot k_s$ dimensions from the dense FFN and directly copy the corresponding weights, or apply Drop-Upcycling to those sampled dimensions. We apply weight scaling to both types of experts.

Note that whether shared experts maintain the same functionality as dense remains an open research question, and comparing initialization methods for shared experts is left for future work.

### C.6.3 LIMITATIONS AND FUTURE DIRECTIONS

While we provide basic extensions of our method to fine-grained and shared expert settings, several important research questions remain unexplored. Our method could serve as a baseline for investigating how knowledge from dense models transfers to these advanced MoE architectures. Specifically, analyzing the transformation process from dense to fine-grained or shared experts could provide valuable insights into how these architectures function and develop specialization. For example, tracking how knowledge is distributed across fine-grained experts during training, or understanding what types of information shared experts learn to capture, could deepen our understanding of these MoE variants. Such analyses could also inform better initialization strategies and architectural choices for future MoE models.

