# OpenReview forum: "Drop-Upcycling: Training Sparse Mixture of Experts with Partial Re-initialization"
_ICLR.cc/2025/Conference — ICLR 2025 Poster_

### Official Review · Reviewer_QL39 · 2024-11-01

**Soundness:** 3
**Presentation:** 3
**Contribution:** 3
**Rating:** 6
**Confidence:** 3

**Summary:**

This paper presents Drop-upcycling, which aims to boost the effectiveness of upcycling method by selectively re-initializing parameters to achieve the trade-off between expert diversity and knowledge transfers. Besides, this work provides a detailed introduction of employing upcycling for the initialization of MoE models, with good readability. The experiments are designed using two dense models such as Llama and Mixtral, and this work includes extensive experimental evaluation across seven types of tasks from both Japanese and English.

**Strengths:**

- 1) This paper is well-written and the Method Section is well-organized, thoroughly demonstrating the authors' understanding of the upcycling methods for MoE models. After reading the Method Section, readers gain valuable insights and can quickly follow the authors' motivation.

- 2) The Drop-upcycling proposed in this paper to enhance the specialization of experts in MoE models is very straightforward, and it includes a hyperparameter that can control the degree of expert specialization and knowledge transfer.

- 3) The experiments in this work are comprehensive, and the Experiment Section provides relatively analyses of the observed phenomena.

**Weaknesses:**

- 1) In Table 1, the training costs of different MoE initialization methods vary. It is recommended to add a comparison of training times for different models to demonstrate the advantages of your proposed method.

- 2) In line 428, the author uses Model 1, Models 2 and 3, etc., to describe the performance comparison of different baseline models. However, it is unclear which model each number represents. It is recommended to add a "Baseline Models" section to introduce the names and methods of different baseline models. The same issue also appears in line 411.

- 3) There are many ways to enhance the specialization of each expert in MoE models. For example, the Branch-Train-MiX (BTX) [Ref A] ensures expert diversity and specialization by feeding each expert different data distributions. In contrast, this work proposes a hyperparameter $r$ to control the initialization density of the FFN layer to ensure specialization. What are the advantages of this approach compared to the BTX method? Please give a brief discussion during the rebuttal period.

[Ref A] Branch-Train-MiX: Mixing Expert LLMs into a Mixture-of-Experts LLM

**Questions:**

- 1) In the caption of Figure 2, please add the definition of $S$ to ensure self-containment.

---

> ### Author Response · Authors · 2024-11-24
>
> We sincerely thank the reviewer for their thorough evaluation and the positive assessment of our paper's clarity, methodology, and experimental comprehensiveness.
>
> ## Clarification on Computational Costs
>
> > In Table 1, the training costs of different MoE initialization methods vary. It is recommended to add a comparison of training times for different models to demonstrate the advantages of your proposed method.
>
> Thank you for your inquiry about training costs. Our proposed method only modifies the initialization of MoE models, which is an extremely lightweight process completing in a matter of seconds. Since the training after initialization follows exactly the same process, our method operates with computational costs equivalent to From Scratch (FS) and Naive Upcycling (NU). Branch-Train-MiX (BTX) requires additional computational costs to create three experts specialized in code, Japanese, and English. We have added the number of training tokens and required FLOPs to Table 1 to directly address this concern and demonstrate computational equivalence.
>
> ## Model Reference Clarification and Baseline Description
>
> > In line 428, the author uses Model 1, Models 2 and 3, etc., to describe the performance comparison of different baseline models. However, it is unclear which model each number represents. It is recommended to add a "Baseline Models" section to introduce the names and methods of different baseline models. The same issue also appears in line 411.
>
> Thank you for pointing out the potential confusion regarding model references. To address this, we have clarified the correspondence between the model numbers and the table by adding the following explanation to both lines 411 and 428:
>
> > Model numbers refer to the leftmost column of this table.
>
> This ensures that readers can easily identify which models are being discussed without ambiguity.
>
> ## Computational Efficiency and Performance Advantages over BTX
>
> > There are many ways to enhance the specialization of each expert in MoE models. For example, the Branch-Train-MiX (BTX) [Ref A] ensures expert diversity and specialization by feeding each expert different data distributions. In contrast, this work proposes a hyperparameter $r$ to control the initialization density of the FFN layer to ensure specialization. What are the advantages of this approach compared to the BTX method? Please give a brief discussion during the rebuttal period.
>
> We appreciate the reviewer's question regarding the advantages of our approach compared to Branch-Train-MiX (BTX).
> Our method achieves superior performance while being more computationally efficient than BTX. As shown in Table 1, our approach using 500B tokens outperforms BTX even when it uses 800B tokens (including 300B additional tokens for expert pre-training). Specifically:
> * **8×152M Model**: Our method achieves a score of **19.7** using $0.91×10^{21}$ FLOPs, compared to BTX 18.5 using $1.39×10^{21}$ FLOPs.
>
> * **8×1.5B Model**: Our method achieves a score of **40.3** using $9.05×10^{21}$ FLOPs, compared to BTX 38.6 using $12.58×10^{21}$ FLOPs.
>
> Beyond computational efficiency, our method offers greater flexibility in training. While BTX enforces expert specialization through predefined data distributions, our approach allows natural specialization to emerge during training through router-expert interactions. This dynamic adaptation leads to more effective learning, as evidenced by consistent performance improvements across model sizes.
> Our findings align with recent research in Skywork-MoE [1], which indicates that BTX's improvements are limited (typically showing loss differences of ~0.01). This further supports the broader applicability and scalability of our approach.
>
> [1] Wei et al. Skywork-MoE: A Deep Dive into Training Techniques for Mixture-of-Experts Language Models. https://arxiv.org/abs/2406.06563
>
> ## Caption Clarification: Definition Addition
> > In the caption of Figure 2, please add the definition of $S$ to ensure self-containment.
>
> We have added the definition of $\mathcal{S}$ to the caption of Figure 2.

---

> > ### Comment · Reviewer_QL39 · 2024-11-25
> > **Thanks for the authors' rebuttal**
> >
> > Thank you for the authors' response, which addressed my first-round concerns. Through the comparison with BTX, I believe this work has certain advantages in terms of computational efficiency and upcycling performance. Therefore, I vote to accept.

---

### Official Review · Reviewer_jqcD · 2024-11-03

**Soundness:** 3
**Presentation:** 2
**Contribution:** 3
**Rating:** 8
**Confidence:** 3

**Summary:**

This paper introduces Drop-Upcycling, a novel method for initializing Mixture of Experts (MoE) models from pre-trained dense models. The key innovation is selectively re-initializing parameters of expert feedforward networks while preserving some knowledge from the pre-trained model. Through extensive experiments, the authors demonstrate that Drop-Upcycling outperforms existing initialization methods like naive Upcycling and Branch-Train-MiX, achieving better performance with reduced training costs. Their best MoE model with 5.9B active parameters matches the performance of a 13B dense model while requiring only 1/4 of the training FLOPs.

**Strengths:**

1. The paper presents a clear technical contribution with Drop-Upcycling that addresses a fundamental challenge in MoE training - balancing knowledge transfer and expert specialization. This is demonstrated through comprehensive empirical results.
2. The experimental evaluation is thorough and well-designed, including ablation studies on re-initialization ratios, detailed analysis of expert routing patterns, and large-scale experiments across multiple model sizes (152M to 3.7B).
3. The work demonstrates significant practical impact by achieving comparable performance to larger dense models with substantially reduced computational costs, which is particularly relevant given the increasing focus on efficient training of large language models.

**Weaknesses:**

1. While the paper shows Drop-Upcycling works well for decoder-only transformers with MoE in FFN layers, there's limited discussion or analysis of how the method might generalize to other architectures (e.g., encoder-decoder models) or different MoE configurations (e.g., attention-based MoE layers).
2. The theoretical analysis in Section 3.2.2 is relatively brief and could benefit from a bit more substance. I understand the space constraints, but it would be great if the authors could include some formal verification or empirical validation for the approximations used in equation (5).
3. The paper uses a fixed re-initialization ratio (r=0.5) across all layers and experts. Given that different layers in transformer models are known to learn different types of features, a more adaptive approach to setting the re-initialization ratio based on layer position or expert functionality might yield better results. The authors don't explore this possibility or justify why a uniform ratio is optimal.

**Questions:**

1. As a suggestion, the authors could consider expanding the theoretical analysis section with formal proofs or empirical validation of the approximations used. This would strengthen the paper's theoretical foundations.
2. It would be valuable to include experiments or discussions on how Drop-Upcycling performs with different MoE architectures or configurations beyond the standard setup used in the paper.
3. Despite the Qwen2 report being brief, the authors could provide a little more detailed analysis comparing their method with Qwen2-MoE,  highlighting any technical differences and their implications. This would be important given that the authors themselves point out the similarity in the two papers. In any case, I appreciate the "open investigation" into this kind of approach.
4. It would be interesting to investigate whether using different re-initialization ratios for different layers (e.g., lower ratios in early layers and higher ratios in later layers) could improve performance. The authors could conduct experiments with layer-wise adaptive ratios or provide an analysis justifying why a uniform ratio is sufficient.
5. It would also be great to see an analysis of the computational overhead introduced by the re-initialization process compared to naive Upcycling.

---

> ### Author Response · Authors · 2024-11-24
> **Response to Reviewer 3 (jqcD) - Part 1**
>
> We sincerely thank the reviewer for their comprehensive assessment of our technical contribution and appreciation of our method's balance between knowledge retention and diversity.
>
> ## Research Scope and Resource Constraints
> **Generalization to Other Architectures and MoE Configurations.**
>
> > While the paper shows Drop-Upcycling works well for decoder-only transformers with MoE in FFN layers, there's limited discussion or analysis of how the method might generalize to other architectures (e.g., encoder-decoder models) or different MoE configurations (e.g., attention-based MoE layers).
>
> Thank you for suggesting experiments to assess the robustness of Drop-Upcycling by applying it to encoder-decoder models and different MoE configurations, such as attention-based MoE layers. We share your interest in understanding the generalization ability of Drop-Upcycling across configurations beyond decoder-only models and the FFN-based MoE approach. However, it is challenging to explore all such combinations within the scope of a single study. In this paper, we have chosen to focus on the most important and widely used configuration—decoder-only models and the FFN-based MoE approach—since conducting all the experiments shown in this paper, including the trial-and-error process, has already taken nearly half a year. We believe that focusing on decoder-only models and the FFN-based MoE approach still offers significant contributions to the community. We consider this suggestion valuable but beyond the scope of the current paper’s focus, categorizing it as a “nice-to-have” suggestion. We kindly ask for the reviewer’s understanding and propose addressing this promising direction in future research.
>
> **Adaptive Re-Initialization Ratios.**
>
> >The paper uses a fixed re-initialization ratio (r=0.5) across all layers and experts. Given that different layers in transformer models are known to learn different types of features, a more adaptive approach to setting the re-initialization ratio based on layer position or expert functionality might yield better results. The authors don't explore this possibility or justify why a uniform ratio is optimal.
>
> > It would be interesting to investigate whether using different re-initialization ratios for different layers (e.g., lower ratios in early layers and higher ratios in later layers) could improve performance. The authors could conduct experiments with layer-wise adaptive ratios or provide an analysis justifying why a uniform ratio is sufficient.
>
> Thank you once again for suggesting such an exciting research direction based on our study. Exploring a dynamic re-initialization ratio that varies depending on the layers is indeed a fascinating idea. For instance, we could set $r$ to range from small to large values, such as 0.25 to 0.75, in accordance with the layer order, as [1] report that parameters in the deeper layers tend to be more diverse than those in the earlier layers. Unfortunately, it would be challenging to conduct experiments and evaluate the results of this approach within the rebuttal period. Nevertheless, we will strive to assess this approach for the final version to enhance our method.
>
> At this stage, we would like to clarify that we never considered the use of a fixed re-initialization ratio (r = 0.5) to be optimal.
> However, it is noteworthy and generally desirable if a straightforward approach performs sufficiently well.
> From this perspective, we can confirm that even a simple method using the fixed re-initialization ratio (r = 0.5) outperforms several baseline methods, as demonstrated in, for example, Tables 1 and 2. These results highlight that the current Drop-Upcycling method already exhibits favorable properties.
>
> [1] Lo et al. A Closer Look into Mixture-of-Experts in Large Language Models.
> https://arxiv.org/abs/2406.18219
>
> ## Enhanced Theoretical Analysis
>
> > The theoretical analysis in Section 3.2.2 is relatively brief and could benefit from a bit more substance. I understand the space constraints, but it would be great if the authors could include some formal verification or empirical validation for the approximations used in equation (5).
>
> > As a suggestion, the authors could consider expanding the theoretical analysis section with formal proofs or empirical validation of the approximations used. This would strengthen the paper's theoretical foundations.
>
> Thank you for this insightful feedback. We acknowledge that Equation (5) in Section 3.2.2 contained an imprecise approximation. We have revised the theoretical analysis:
>
> 1. We have corrected and enhanced Section 3.2.2 with a more precise formulation of the parameter sharing characteristics.
> 2. We have added a detailed derivation in Appendix C.5 that formally validates the approximations.
>
> These revisions provide a more rigorous theoretical foundation. We have carefully verified that this mathematical clarification is purely explanatory and does not affect any experimental results or conclusions.

---

> ### Author Response · Authors · 2024-11-24
> **Response to Reviewer 3 (jqcD) - Part 2**
>
> ## Advanced MoE Architectures
>
> > It would be valuable to include experiments or discussions on how Drop-Upcycling performs with different MoE architectures or configurations beyond the standard setup used in the paper.
>
> We add a detailed discussion in Appendix C.6 about extending our method to fine-grained MoEs and shared expert designs. Our analysis shows that Drop-Upcycling can be naturally extended to fine-grained experts by applying partial re-initialization with ratio $r$, though weights need to be scaled to account for the increased number of activated experts. For shared experts, while both preserving the original parameters and applying Drop-Upcycling are possible approaches, determining the optimal initialization strategy remains an open question for future research. We hope these initial extensions can serve as a foundation for further investigations into advanced MoE architectures.
>
> ## Relationship to Qwen2
>
> > Despite the Qwen2 report being brief, the authors could provide a little more detailed analysis comparing their method with Qwen2-MoE, highlighting any technical differences and their implications. This would be important given that the authors themselves point out the similarity in the two papers. In any case, I appreciate the "open investigation" into this kind of approach.
>
> We appreciate your suggestion to highlight the difference and comparison with Qwen2. The Qwen2 technical report lacks detailed procedures for reproducing the Qwen2-MoE training, and the authors have not yet released any code for the MoE training. This makes it hard to reimplement the Qwen2-MoE training process.
>
> However, in order to comply with the reviewers' requests to the best of our ability, we tried to reproduce the Qwen2-MoE experiments using settings we could understand, while incorporating typical configurations to account for unknown elements. As far as we understand, Drop-Upcycling differs from Qwen2-MoE in the initialization method: Drop-Upcycling uses random initialization to replace the trained weights of the dense model, whereas Qwen2-MoE applies noise addition to the trained weights. As noted in the OLMoE [1] paper (Appendix F, "Noise Upcycling"), “For the creation of Qwen2-MoE [200, 178, 13], the authors add 50% of Gaussian noise to feedforward networks before continuing training in an upcycled setup [84].” Following this description, we conducted additional experiments with our 8×152M model, randomly selecting 50% of the parameters and adding Gaussian noise with standard deviation 0.02 to these selected parameters, to directly compare these initialization strategies.
>
> Our experimental results for the 8×152M model show the following performance across evaluation tasks:
>
> | Method | JEMHQA | NIILC | JSQ | XL-Sum | WMT E→J | WMT J→E | OBQA | TQA | HS | SQv2 | XW-EN | BBH | Avg |
> |--------|---------|--------|-----|---------|----------|----------|-------|-----|-----|-------|--------|-----|-----|
> | Dense | 17.6 | 7.9 | 10.6 | 2.4 | 0.5 | 0.5 | 14.6 | 3.0 | 28.6 | 2.0 | 60.6 | 11.5 | 13.3 |
> | From Scratch | 25.2 | 13.6 | 19.4 | 1.8 | 0.9 | 0.4 | 16.6 | 2.6 | 31.2 | **12.9** | 64.4 | 10.7 | 16.6 |
> | BTX | 28.6 | 17.1 | 26.6 | **4.3** | 2.7 | 1.1 | **18.4** | 5.1 | **32.5** | 5.3 | **65.0** | 15.9 | 18.5 |
> | Naive Upcycling | 28.2 | 16.2 | 24.4 | 3.5 | 3.0 | 1.1 | 18.2 | 5.8 | 31.9 | 4.5 | 63.5 | 14.7 | 17.9 |
> | Noise Upcycling | 28.6 | 17.1 | 29.4 | 3.7 | 2.3 | 1.6 | 16.8 | 5.3 | 32.0 | 4.8 | 64.5 | 17.4 | 18.6 |
> | Drop-Upcycling (r=0.5)| **32.2** | **18.0** | 30.6 | 3.7 | **4.7** | **2.3** | 16.8 | **6.1** | **32.5** | 6.2 | 64.2 | **19.1** | **19.7** |
> | Drop-Upcycling (r=1.0) | 27.2 | 16.8 | **32.5** | 4.1 | 3.7 | 1.6 | 17.0 | 5.9 | 32.4 | 4.9 | 64.8 | 15.4 | 18.9 |
>
> Our experiments show that our method outperforms Noise Upcycling. While we have completed all experiments with the 8×152M model, the 8×1.5B experiments are ongoing, with current results and training curves included in the updated supplementary material ZIP file ( as`noise_upcycling_baseline_exp.pdf`). Complete results will be included in the camera-ready version
>
> [1] Muennighoff et al. OLMoE: Open Mixture-of-Experts Language Models.
> https://arxiv.org/abs/2409.02060
>
>
>
>
> ## Clarification on Computational Costs
>
> > It would also be great to see an analysis of the computational overhead introduced by the re-initialization process compared to naive Upcycling.
>
> Our proposed method only modifies the initialization of MoE models, which is an extremely lightweight process completing in a matter of seconds. Since the training after initialization follows exactly the same process, the computational overhead introduced by our method is negligible. Therefore, the training costs for our proposed method and the baselines are identical, except for Branch-Train-MiX (BTX), which incurs additional costs due to the separate training of experts. We updated Table 1 to include training tokens and FLOPs to clarify this point.

---

> > ### Comment · Reviewer_jqcD · 2024-11-26
> > **Thanks for the rebuttal**
> >
> > I thank the authors for their comprehensive answers to my queries. I also acknowledge that they are limited by the unavailability of code for reproducing certain experiments on Qwen2. In any case, the rebuttal clarified a few of my queries about the paper and laid the path for some interesting future works. I have already voted to accept it, and the discussion has increased my confidence in that rating.

---

### Official Review · Reviewer_u1Mr · 2024-11-03

**Soundness:** 3
**Presentation:** 3
**Contribution:** 2
**Rating:** 6
**Confidence:** 4

**Summary:**

This paper proposes "Drop-Upcycling",  a method that enables an improved initialization scheme for expert parameters based on the weights of a pre-trained dense model. The approach addresses the lack of diversity in expert weights, a limitation of the standard Upcycling method where expert weights are simply cloned from the FFN in the dense model.

Drop-Upcycling randomly samples indices from the intermediate dimension of the FFNs. The weights corresponding to these indices from $W_{up}$, $W_{gate}$, and $W_{down}$ matrices are dropped and reinitialized using the statistics of the dropped weights. The parameter $ r $ controls the proportion of indices sampled, with $ r = 1 $ representing full random initialization and $ r = 0 $ corresponding to naive upcycling.

The authors performed thorough experiments with an overall 200k plus GPU hours. A key insight is that the advantages of the approach is mainly in long-term training scenarios (>100B tokens).

**Strengths:**

- The empirical evaluation section and the number of large-scale training conducted are impressive.
- The proposed method is very simple and the parameter $r$ intuitively controls the trade-off between knowledge retention from the dense network and incorporating diversity.
- The authors implement their drop-upcycling on different parameter scale regimes from ~400M all the way to 18B parameters.
- The authors provide the source code which is great for reproducibility

**Weaknesses:**

- Although the empirical evaluation is thorough and the results are promising, the paper has rather limited novelty. The paper would have become stronger if the authors could handle more challenging cases appearing in recent MoE literature such as fine-grained MoEs, or experimenting with a shared expert design. It would be beneficial to report the result of some other intuitive baselines such as adding random noise to the channels of the upcycled experts.

- The way the load-balancing loss is applied seems ineffective. Could you elaborate why the loss was not applied layer-wise as that should prevent expert collapse. Even the model trained with Drop-upcycling shows half of the experts in layer 23 are not used by any of the domains.

- The authors mention that the proposed method outperforms other upcycling approaches in long-term training scenarios. It seems in the very long-term training setting, training from scratch may outperform all methods and in the very short-term training (e.g. <50B tokens) the Branch-Train-Mix upcycling achieves the lowest loss. It would be best if the authors could explain this angle better and also study effect of model parameter size. For example at r=0.5, it takes around 70B Tokens to surpass Branch-Train-Mix for the $8\times 152M$ model, while ~130B tokens are required for surpassing it for the $8\times 1.5B$ Tokens.

**Questions:**

- As the authors highlighted, the Qwen1.5-2.7B-MoE model seems to follow a similar initialization strategy as Drop-Upcycling. It would be interesting to see how this initialization strategy could be extended for fine-grained MoEs with Granularity >1. Do the authors have thought on how they could extend their method accordingly?

---

> ### Author Response · Authors · 2024-11-24
> **Response to Reviewer 2 (u1Mr) - Part 1**
>
> We sincerely thank the reviewer for their detailed assessment and recognition of our extensive experimental evaluation. We appreciate the thorough feedback highlighting both our empirical contributions and areas for potential improvement.
>
> ## Extension to Advanced MoE Architectures and Additional Baseline Comparisons
>
> > Although the empirical evaluation is thorough and the results are promising, the paper has rather limited novelty. The paper would have become stronger if the authors could handle more challenging cases appearing in recent MoE literature such as fine-grained MoEs, or experimenting with a shared expert design. It would be beneficial to report the result of some other intuitive baselines such as adding random noise to the channels of the upcycled experts.
>
> > As the authors highlighted, the Qwen1.5-2.7B-MoE model seems to follow a similar initialization strategy as Drop-Upcycling. It would be interesting to see how this initialization strategy could be extended for fine-grained MoEs with Granularity >1. Do the authors have thought on how they could extend their method accordingly?
>
> **Advanced MoE Architectures**
>
> We add a detailed discussion in Appendix C.6 about extending our method to fine-grained MoEs and shared expert designs. Our analysis shows that Drop-Upcycling can be naturally extended to fine-grained experts by applying partial re-initialization with ratio $r$, though weights need to be scaled to account for the increased number of activated experts. For shared experts, while both preserving the original parameters and applying Drop-Upcycling are possible approaches, determining the optimal initialization strategy remains an open question for future research. We hope these initial extensions can serve as a foundation for further investigations into advanced MoE architectures.
>
> **Experimental Comparison**
>
> In response to the reviewers' request, we have added a comparison with a baseline that involves adding random noise. We refer to this approach as Noise Upcycling. We conducted additional experiments with our 8×152M model, randomly selecting 50% of the parameters and adding Gaussian noise with standard deviation 0.02 to these selected parameters, to directly compare these initialization strategies.
>
> Our experimental results for the 8×152M model show the following performance across evaluation tasks:
>
> | Method | JEMHQA | NIILC | JSQ | XL-Sum | WMT E→J | WMT J→E | OBQA | TQA | HS | SQv2 | XW-EN | BBH | Avg |
> |--------|---------|--------|-----|---------|----------|----------|-------|-----|-----|-------|--------|-----|-----|
> | Dense | 17.6 | 7.9 | 10.6 | 2.4 | 0.5 | 0.5 | 14.6 | 3.0 | 28.6 | 2.0 | 60.6 | 11.5 | 13.3 |
> | From Scratch | 25.2 | 13.6 | 19.4 | 1.8 | 0.9 | 0.4 | 16.6 | 2.6 | 31.2 | **12.9** | 64.4 | 10.7 | 16.6 |
> | BTX | 28.6 | 17.1 | 26.6 | **4.3** | 2.7 | 1.1 | **18.4** | 5.1 | **32.5** | 5.3 | **65.0** | 15.9 | 18.5 |
> | Naive Upcycling | 28.2 | 16.2 | 24.4 | 3.5 | 3.0 | 1.1 | 18.2 | 5.8 | 31.9 | 4.5 | 63.5 | 14.7 | 17.9 |
> | Noise Upcycling | 28.6 | 17.1 | 29.4 | 3.7 | 2.3 | 1.6 | 16.8 | 5.3 | 32.0 | 4.8 | 64.5 | 17.4 | 18.6 |
> | Drop-Upcycling (r=0.5)| **32.2** | **18.0** | 30.6 | 3.7 | **4.7** | **2.3** | 16.8 | **6.1** | **32.5** | 6.2 | 64.2 | **19.1** | **19.7** |
> | Drop-Upcycling (r=1.0) | 27.2 | 16.8 | **32.5** | 4.1 | 3.7 | 1.6 | 17.0 | 5.9 | 32.4 | 4.9 | 64.8 | 15.4 | 18.9 |
>
> Our experiments show that our method outperforms Noise Upcycling. While we have completed all experiments with the 8×152M model, the 8×1.5B experiments are ongoing, with current results and training curves included in the updated supplementary material ZIP file (as `noise_upcycling_baseline_exp.pdf`). Complete results will be included in the camera-ready version.

---

> ### Author Response · Authors · 2024-11-24
> **Response to Reviewer 2 (u1Mr) - Part 2**
>
> ## Comparative Analysis of Global vs Layer-wise Load Balancing
>
> > The way the load-balancing loss is applied seems ineffective. Could you elaborate why the loss was not applied layer-wise as that should prevent expert collapse. Even the model trained with Drop-upcycling shows half of the experts in layer 23 are not used by any of the domains.
>
> We appreciate the reviewer’s observation regarding the potential benefits of layer-wise load balancing. In our experiments, we applied the load balancing loss globally rather than layer-wise. This approach aligns with standard practices in common implementations (e.g., HuggingFace transformers).
>
> To address this concern, we have added a comprehensive analysis of the effect of global and layer-wise load balancing in Appendix C.3 that includes detailed loss trajectories and evaluation scores for various initialization methods (Naive Upcycling, From Scratch, Branch-Train-MiX, Drop-Upcycle (r=0.5, r=1.0)) in the 8×1.5B setting. Our findings reveal that there are no significant differences in both loss and evaluation scores between global and layer-wise load balancing across these methods.
>
> Furthermore, our analysis demonstrates that Drop-Upcycling, similar to training from scratch, maintains domain specialization across most layers even when applying layer-wise load balancing.
>
> The trade-offs between layer-wise and global load balancing, alongside broader questions about expert utilization and architectural choices (e.g., varying expert counts per layer), remain compelling directions for future research. We believe these aspects of MoE design warrant further investigation and could lead to more efficient architectures.
> ## Performance Across Training Regimes
> > The authors mention that the proposed method outperforms other upcycling approaches in long-term training scenarios. It seems in the very long-term training setting, training from scratch may outperform all methods and in the very short-term training (e.g. <50B tokens) the Branch-Train-Mix upcycling achieves the lowest loss. It would be best if the authors could explain this angle better and also study effect of model parameter size. For example at r=0.5, it takes around 70B Tokens to surpass Branch-Train-Mix for the $8\times152M$ model, while ~130B tokens are required for surpassing it for the $8\times1.5B$ Tokens.
>
> Thank you for your suggestion to explore and extend the message of our paper further. In response, we have added an analysis to Appendix C.4 to address this point. As noted in the appendix, while this study is not without limitations, such as the impact of LR scheduling, we believe it provides valuable insights. We encourage you to refer to the appendix for further details.
>
> Specifically, we observed no tendency for training from scratch to catch up with Drop-Upcycling. This shows that training from scratch cannot match Drop-Upcycling unless an impractically large budget is allocated, underscoring Drop-Upcycling as the better option in practical scenarios. For a further related discussion on why upcycling-based approaches present a practical alternative, we also recommend referring to [1, 2].
>
> Additionally, we would like to highlight that when comparing Drop-Upcycling against Branch-Train-Mix (BTX), it is important to account for the additional budget spent on training individual experts. BTX uses a significant extra budget for expert training before MoE training, which should be factored into any direct comparison.
>
> We appreciate the reviewer’s insightful comments, which motivated this additional study.
>
> [1] Komatsuzaki et al. Sparse Upcycling: Training Mixture-of-Experts from Dense Checkpoints. ICLR’23. https://arxiv.org/abs/2212.05055
>
> [2] Wei et al. Skywork-MoE: A Deep Dive into Training Techniques for Mixture-of-Experts Language Models. https://arxiv.org/abs/2406.06563

---

> > ### Author Response · Authors · 2024-11-29
> > **Reminder**
> >
> > Dear Reviewer u1Mr,
> >
> > As the discussion period is nearing its end, we kindly remind you to review our response if you haven't had the chance yet. We are keen to know if our response has addressed your concerns and would be grateful if you could reconsider the rating if appropriate. If there are any further questions or clarifications needed, we would be more than happy to provide additional information.
> >
> > Thank you very much for your time and consideration.
> >
> > Authors

---

> > ### Comment · Reviewer_u1Mr · 2024-11-30
> >
> > I would like to thank the authors for their thorough responses and the added experiments and descriptions to the paper. They have largely addressed my concerns and I therefore raise my score to 6.

---

### Official Review · Reviewer_FeSK · 2024-11-12

**Soundness:** 2
**Presentation:** 3
**Contribution:** 2
**Rating:** 5
**Confidence:** 4

**Summary:**

This paper presents Drop-Upcycling for MoE training utilizing the knowledge of pre-trained dense models while statistically re-initializing
some parts of the weights. The proposed method promotes expert specialization (which is often a bottleneck in traditional upcycling), significantly enhancing the MoE model’s efficiency in knowledge acquisition.

**Strengths:**

- The paper is well written and easy to follow.
- The idea of drop-upcycling is interesting although not new (see weaknesses section).
- Extensive experiments and release of all the relevant materials for reproducibility.

**Weaknesses:**

- How is the proposed upcycling method different from the upcycling used in Qwen2: https://arxiv.org/pdf/2407.10671? Qwen2 shuffles parameters along the intermediate dimension to promote diversity which I think is very similar to the current method. They also randomly initialize 50% of the parameters. I would suggest the authors to discuss the differences with experiments.
- What are the benefits of upcycling over training from scratch? I think changing architectural components are often difficult as upcycling is often constrained by the dense model architecture.
- Will the current method going to be scalable to very large MoEs, say 100B+ parameter models? What additional challenges author think to encounter in training large MoEs with upcycling?
- Can upcycling going to help in continual learning? E.g., can we add new experts to an existing MoE using dropupcycling?

**Questions:**

I think the identified problem is important but I’d like to rate the current submission below the acceptance threshold (included towards reject) due to limited technical contributions, mainly similarity with existing upcycling methods like the one used in Qwen2.

---

> ### Author Response · Authors · 2024-11-24
> **Response to Reviewer 1 (FeSK) - Part 1**
>
> We sincerely thank the reviewer for their detailed feedback and recognition of our paper's clarity and experimental thoroughness.
>
> ## Relationship to Qwen2
> > How is the proposed upcycling method different from the upcycling used in Qwen2: https://arxiv.org/pdf/2407.10671? Qwen2 shuffles parameters along the intermediate dimension to promote diversity which I think is very similar to the current method. They also randomly initialize 50% of the parameters. I would suggest the authors to discuss the differences with experiments.
>
> > I think the identified problem is important but I’d like to rate the current submission below the acceptance threshold (included towards reject) due to limited technical contributions, mainly similarity with existing upcycling methods like the one used in Qwen2.
>
> **Research Originality**
>
> Thank you for highlighting the similarity between the proposed method, Drop-Upcycling, and Qwen2-MoE. We have already discussed Qwen2-MoE in detail in the Related Work section of the submission. Please refer to Section 2.2 (line 149) for more information.
>
> > L149: Concurrent with our work, the Qwen2 technical report  (Yang et al., 2024) briefly suggests the use of a methodology possibly related to Drop-Upcycling in training Qwen2-MoE. Due to the report's brevity and ambiguity, it is unclear if their method exactly matches ours. Our paper offers a valuable technical contribution even if the methods are similar.  The potential application of Drop-Upcycling in an advanced, industry-developed model like Qwen2-MoE that underscores the importance of further open investigation into this approach. We acknowledge the Qwen2 authors for sharing insights through their technical report.
>
> We attempted to reimplement the Qwen2-MoE training process but were unable to do so. This is because the Qwen2 technical report lacks detailed procedures for reproducing the Qwen2-MoE training, and the authors have not yet released any code for the MoE training.
>
> This is one of the reasons why we included the statement, “All experimental resources, including source code, training data, model checkpoints and logs, are publicly available to promote reproducibility and future research on MoE,” in our abstract.
>
> From another perspective, we would like to emphasize that our work and Qwen2-MoE should be considered concurrent works, as the Qwen2 paper was released in July 2024, less than 3 months before our submission.
>
> Based on the reasons outlined above, we believe that the absence of a direct comparison between Qwen2-MoE and Drop-Upcycling in our experiments does not constitute a significant reason for rejection or a notable weakness of our paper.

---

> ### Author Response · Authors · 2024-11-24
> **Response to Reviewer 1 (FeSK) - Part 2**
>
> **Experimental Comparison.**
>
> However, in order to comply with the reviewers' requests to the best of our ability, we tried to reproduce the Qwen2-MoE experiments using settings we could understand, while incorporating typical configurations to account for unknown elements. As far as we understand, Drop-Upcycling differs from Qwen2-MoE in the initialization method: Drop-Upcycling uses random initialization to replace the trained weights of the dense model, whereas Qwen2-MoE applies noise addition to the trained weights. As noted in the OLMoE [1] paper (Appendix F, "Noise Upcycling"), “For the creation of Qwen2-MoE [200, 178, 13], the authors add 50% of Gaussian noise to feedforward networks before continuing training in an upcycled setup [84].” Following this description, we conducted additional experiments with our 8×152M model, randomly selecting 50% of the parameters and adding Gaussian noise with standard deviation 0.02 to these selected parameters, to directly compare these initialization strategies.
>
> Our experimental results for the 8×152M model show the following performance across evaluation tasks:
>
> | Method | JEMHQA | NIILC | JSQ | XL-Sum | WMT E→J | WMT J→E | OBQA | TQA | HS | SQv2 | XW-EN | BBH | Avg |
> |--------|---------|--------|-----|---------|----------|----------|-------|-----|-----|-------|--------|-----|-----|
> | Dense | 17.6 | 7.9 | 10.6 | 2.4 | 0.5 | 0.5 | 14.6 | 3.0 | 28.6 | 2.0 | 60.6 | 11.5 | 13.3 |
> | From Scratch | 25.2 | 13.6 | 19.4 | 1.8 | 0.9 | 0.4 | 16.6 | 2.6 | 31.2 | **12.9** | 64.4 | 10.7 | 16.6 |
> | BTX | 28.6 | 17.1 | 26.6 | **4.3** | 2.7 | 1.1 | **18.4** | 5.1 | **32.5** | 5.3 | **65.0** | 15.9 | 18.5 |
> | Naive Upcycling | 28.2 | 16.2 | 24.4 | 3.5 | 3.0 | 1.1 | 18.2 | 5.8 | 31.9 | 4.5 | 63.5 | 14.7 | 17.9 |
> | Noise Upcycling | 28.6 | 17.1 | 29.4 | 3.7 | 2.3 | 1.6 | 16.8 | 5.3 | 32.0 | 4.8 | 64.5 | 17.4 | 18.6 |
> | Drop-Upcycling (r=0.5)| **32.2** | **18.0** | 30.6 | 3.7 | **4.7** | **2.3** | 16.8 | **6.1** | **32.5** | 6.2 | 64.2 | **19.1** | **19.7** |
> | Drop-Upcycling (r=1.0) | 27.2 | 16.8 | **32.5** | 4.1 | 3.7 | 1.6 | 17.0 | 5.9 | 32.4 | 4.9 | 64.8 | 15.4 | 18.9 |
>
> Our experiments show that our method outperforms Noise Upcycling. While we have completed all experiments with the 8×152M model, the 8×1.5B experiments are ongoing, with current results and training curves included in the updated supplementary material ZIP file (as `noise_upcycling_baseline_exp.pdf`). Complete results will be included in the camera-ready version.
>
> [1] Muennighoff et al. OLMoE: Open Mixture-of-Experts Language Models.
> https://arxiv.org/abs/2409.02060
>
> ## Benefits of Upcycling over Training from Scratch
> > What are the benefits of upcycling over training from scratch? I think changing architectural components are often difficult as upcycling is often constrained by the dense model architecture.
>
> (It seems that this statement is more of a question than a weakness. If our understanding is correct, we kindly request the reviewer to move this statement to the question section to ensure a fair assessment of our paper.)
>
> The benefit of upcycling over training from scratch is that it helps to reduce the cost of building MoE models or enables better models to be built at the same cost. For example, a well-known open-weight MoE model, Mixtral, may have utilized upcycling techniques. The benefits of upcycling are widely recognized and have already been discussed in several previous papers that propose or use the upcycling technique in the MoE literature. For instance, the Sparse Upcycling [1] paper demonstrated that when the compute budget is limited, their upcycling approach achieves superior performance compared to training from scratch. Similarly, the Branch-Train-MiX [2] paper showed that using dense models as seeds for expert training followed by upcycling leads to better performance than Sparse Upcycling.
>
> Similar to these papers, we also demonstrate the advantages of the upcycling technique in terms of the relationship between the performance achieved in each MoE training approach and its computational cost. Table 1, for example, shows the superior performance of Drop-Upcycling, as indicated in the row “MoE DU (r=0.5),” compared to training from scratch, shown in the row “MoE FS.”
>
> [1] Komatsuzaki et al. Sparse Upcycling: Training Mixture-of-Experts from Dense Checkpoints. ICLR’23. https://arxiv.org/abs/2212.05055
>
> [2] Sukhbaata et al. Branch-Train-MiX: Mixing Expert LLMs into a Mixture-of-Experts LLM. COLM’24
> https://arxiv.org/abs/2403.07816

---

> ### Author Response · Authors · 2024-11-24
> **Response to Reviewer 1 (FeSK) - Part 3**
>
> ## Scalability to Larger Models
> > Will the current method going to be scalable to very large MoEs, say 100B+ parameter models? What additional challenges author think to encounter in training large MoEs with upcycling?
>
> (Same as the above, it seems that this statement is more of a question than a weakness. If our understanding is correct, we kindly request the reviewer to move this statement to the question section to ensure a fair assessment of our paper.)
>
> As discussed in Section 5.2, we evaluated various model sizes (e.g., 8x152M, 8x1.5B, and 8x3.7B) and consistently observed performance improvements as the model size increased (Please compare the performance results in row 5 of MoE DU (r=0.5) in Table 1, row 11 of MoE DU (r=0.5) in Table 1, and row 3 of MoE DU (r=0.5) in Table 2). We have no reason to believe that this tendency will change for  100B+ MoE settings. A potential additional challenge of the 100B+ MoE setting is the need for more careful tuning of initialization in terms of both scale and ratio, as larger models have an extremely large number of parameters in the FFN.
>
> Currently, we are testing 10B+ MoE settings, and a 100B+ MoE setting is planned for the next phase. However, we kindly ask the reviewer to consider that conducting a single MoE experiment in a 100B+ MoE setting involves significant computational costs that make such experiments prohibitively expensive for most research teams.
> ## Potential for Continual Learning
> > Can upcycling going to help in continual learning? E.g., can we add new experts to an existing MoE using dropupcycling?
>
> Thank you for providing such an interesting research direction based on our study.  In our view, the concept of Drop-Upcycling could also be beneficial in a continual learning setting, though it may require modifications to adapt to this context. However, we believe that this direction lies beyond the scope of the current paper.  We kindly request the reviewer’s understanding in allowing us to leave it for future research; we will explore your suggested direction in our next study."

---

> > ### Author Response · Authors · 2024-11-29
> > **Reminder**
> >
> > Dear Reviewer FeSK,
> >
> > As the discussion period is nearing its end, we kindly remind you to review our response if you haven't had the chance yet. We are keen to know if our response has addressed your concerns and would be grateful if you could reconsider the rating if appropriate. If there are any further questions or clarifications needed, we would be more than happy to provide additional information.
> >
> > Thank you very much for your time and consideration.
> >
> > Authors

---

### Author Response · Authors · 2024-11-24
**General Response to Reviewers and AC**

We sincerely thank all the reviewers for their time and effort in evaluating our paper, as well as for their constructive and insightful feedback. We have carefully considered the comments and questions provided, and have made revisions to address these concerns. Additionally, we have conducted new experiments and analyses. (All changes are highlighted in purple text in the revised PDF).

**Writing:**
1. Revised theoretical characteristics formulation (see Section 3.2.2)
2. Added detailed derivations of theoretical characteristics (see Appendix C.5)
3. Added analysis of load balancing loss applied at both global and layer-wise levels (see Appendix C.3)
4. Added convergence catch-up analysis (see Appendix C.4)
5. Added discussion on applying our method to fine-grained and shared experts MoE (see Appendix C.6)
6. Updated the experimental results to include training tokens and FLOPs (see Table 1, Section 5)
7. Clarified the correspondence of model numbers in 5.1 METHOD COMPARISON by adding:
Model numbers refer to the leftmost column of this table (see lines 411 and 428).
8. Updated the caption of Figure 2 to include the definition of $\mathcal{S}$, ensuring self-containment.

**Additional Experiments:**
* We have investigated the effects of global load balance vs. layer-wise load balance
* We have also conducted additional experiments with another baseline, Noise Upcycling:
  - Completed for the 8×152M model
  - Preliminary results are presented for the 8×1.5B model, with completion expected by the camera-ready deadline.
* We have updated our supplementary materials with these new experimental results.

We hope that our responses and revisions satisfactorily address the reviewers' concerns. If so, we kindly ask the reviewers to consider re-evaluating their ratings in light of these improvements.

---

### Author Response · Authors · 2024-11-28
**General Response to Reviewers and AC**

The previously announced experiments on Noise Upcycling (which we believe to be the approach used in Qwen2-MoE) with the 8×1.5B model have been completed. Together with the previously shared 8×152M Noise Upcycling results, we updated our manuscript to incorporate these results. All changes are highlighted in purple text in the revised PDF.


**Writing Updates:**
- Added Noise Upcycling baseline settings in Section 4 (We decided to refer to it as "Random Noise Upcycling" in the main text as we realized that the abbreviation NU would overlap with naive Upcycling.)
- Updated Section 5 Table 1 and Figure 3 with the new experimental results


In summary, we confirmed that Drop-Upcycling outperforms Noise Upcycling in both 8×152M and 8×1.5B settings. The following is our experimental results for the 8×1.5B model. For a detailed analysis, including training curves, we encourage readers to refer to Section 5, Table 1, and Figure 3.

| Method | JEMHQA | NIILC | JSQ | XL-Sum | WMT E→J | WMT J→E | OBQA | TQA | HS | SQv2 | XW-EN | BBH | Avg |
|--------|---------|--------|-----|---------|----------|----------|-------|-----|-----|-------|--------|-----|-----|
| Dense | 49.6 | 42.5 | 48.1 | 11.3 | 16.8 | 8.5 | 22.2 | 23.8 | 42.9 | 16.2 | 82.5 | 25.1 | 32.5 |
| From Scratch | 48.3 | 45.4 | 59.1 | 7.5 | 16.6 | 6.9 | 26.4 | 31.5 | 47.3 | 15.0 | 83.7 | 25.9 | 34.5 |
| BTX | 44.3 | 51.8 | 69.4 | 11.9 | 22.4 | **12.5** | 27.8 | 39.2 | 49.7 | 18.7 | **86.4** | 28.9 | 38.6 |
| Naive Upcycling | 50.4 | 50.6 | 61.7 | 12.4 | 21.6 | 10.5 | 26.8 | 36.2 | 47.7 | 19.0 | 85.0 | 27.2 | 37.4 |
| Noise Upcycling | **53.6** | 50.5 | 71.2 | 12.3 | 22.3 | 11.7 | 26.4 | 40.0 | 49.9 | 19.1 | 84.9 | 27.5 | 39.1 |
| Drop-Upcycling (r=0.5) | 51.1 | **52.3** | **72.5** | **13.7** | **22.5** | **12.5** | **30.6** | **41.3** | **50.4** | **21.2** | 86.2 | **29.1** | **40.3** |
| Drop-Upcycling (r=1.0) | 52.1 | 50.9 | 68.8 | 12.3 | 21.9 | 12.4 | 25.0 | 39.1 | 49.7 | 20.6 | 86.0 | 27.9 | 38.9 |



We sincerely thank all reviewers for their constructive feedback. We hope that our responses and revisions satisfactorily address the reviewers' concerns. If so, we kindly ask the reviewers to consider re-evaluating their ratings in light of these improvements.

---

### Meta-Review · Area_Chair_5rbF · 2024-12-24

**Metareview:**

This paper proposes Drop-upcycling, which selectively initializes MoE models from pre-trained dense models, and successfully balances expert diversity and knowledge transfer. The paper is well-written, with a clear and easily understandable motivation, and the experimental section provides extensive and solid evaluation. The reviewers raised concerns regarding similar work and some experimental analyses, which the authors addressed with detailed analysis and additional experiments that demonstrate the superiority of Drop-upcycling, largely alleviating the reviewers' concerns. Overall, this work is good and solid, so the AC recommends accept.

**Additional Comments On Reviewer Discussion:**

The main concern raised by the reviewers was the similarity in the initialization strategy between Drop-Upcycling and the Qwen-MoE model. Since Qwen lacks detailed reproduction procedures and MoE training code, the authors reproduced it based on their own understanding and supplemented the experiments, which partially addressed this issue.

---

### Decision · Program_Chairs · 2025-01-22

Accept (Poster)